# CROCS: Clustering and Retrieval of Cardiac Signals Based on Patient Disease Class, Sex, and Age

**Dani Kiyasseh, Tingting Zhu, & David Clifton**
Department of Engineering Science
University of Oxford
Oxford, UK
`{dani.kiyasseh,tingting.zhu,david.clifton}@eng.ox.ac.uk`

## Abstract

The process of manually searching for relevant instances in, and extracting information from, clinical databases underpin a multitude of clinical tasks. Such tasks include disease diagnosis, clinical trial recruitment, and continuing medical education. This manual search-and-extract process, however, has been hampered by the growth of large-scale clinical databases and the increased prevalence of unlabelled instances. To address this challenge, we propose a supervised contrastive learning framework, CROCS, where representations of cardiac signals associated with a set of patient-specific attributes (e.g., disease class, sex, age) are attracted to learnable embeddings entitled clinical prototypes. We exploit such prototypes for *both* the clustering and retrieval of unlabelled cardiac signals based on multiple patient attributes. We show that CROCS outperforms the state-of-the-art method, DTC, when clustering and also retrieves relevant cardiac signals from a large database. We also show that clinical prototypes adopt a semantically meaningful arrangement based on patient attributes and thus confer a high degree of interpretability.

## 1 Introduction

Clinical databases can comprise instances that are either unlabelled or labelled with patient attribute information, such as disease class, sex, and age. The process of manually searching for relevant instances in, and extracting information from, such databases underpin a multitude of tasks [1]. For example, clinicians extract a disease diagnosis from patient data, researchers involved in clinical trials search for and recruit patients satisfying specific inclusion criteria [2], and educators retrieve relevant information as part of the continuing medical education scheme [3]. This manual search-and-extract process, however, is hampered by the rapid growth of large-scale clinical databases and the increased prevalence of unlabelled instances; those for which patient attribute information is unavailable.

Given such a setting, in this paper, we are interested in addressing two questions: given a large, unlabelled clinical database, (1) *how do we extract attribute information from such unlabelled instances?* and (2) *how do we reliably search for and retrieve relevant instances?* To address the former, the task of clustering holds value. In this setting, a centroid groups together instances that share some similarities. Recent research has focused on exploiting existing clustering algorithms, such as $k$-means, to group similar patients from electronic heath record (EHR) data [4, 5]. Such methods, however, are exclusively unsupervised; they do not exploit patient attribute information. To address the second question, the task of information retrieval holds promise. In this setting, a query associated with a set of desired attributes is exploited to retrieve a relevant instance. Recent research has focused predominantly on retrieving medical images [6], clinical text [7], and EHR data [8], with minimal emphasis on medical time-series data [9]. These methods do not extend to cardiac

time-series data nor do they account for search based on multiple patient attributes. Most notably, previous work performs either clustering or retrieval, and not both.

In this work, we address both questions while exploiting large-scale electrocardiogram (ECG) databases comprising patient attribute information. Our contributions are the following: (1) we propose a supervised contrastive learning framework, CROCS, in which we attract representations of cardiac signals associated with a unique set of patient attributes to embeddings, entitled clinical prototypes. Such attribute-specific prototypes, which create "islands" of similar representations [10], allow for *both* the clustering and retrieval of cardiac signals based on *multiple* patient attributes. (2) We show that CROCS outperforms the state-of-the-art method, DTC, in the clustering setting and retrieves relevant cardiac signals from a large database. At the same time, clinical prototypes adopt a semantically meaningful arrangement and thus confer a high degree of interpretability.

## 2 Related work

**Clinical representation learning and clustering**   Learning meaningful representations of clinical data is an ongoing research endeavour. Recent research has focused on learning representations from EHR data [11, 12, 13, 14, 15] and via auto-encoders, which are then clustered using existing methods, such as $k$-means [4, 5]. As for time-series data, auto-encoders are learned with [16] or without [17] an auxiliary clustering objective, salient features (shapelets) are identified in an unsupervised manner [18, 19], and patient-specific representations are learned via contrastive learning [20]. Li *et al.* [21] learn prototypes, or representative embeddings, via the ProtoNCE loss and cluster instances using $k$-means. Their work builds upon recent research in the contrastive learning literature [22, 23, 24]. Similar to our work is that of Gee *et al.* [11] where prototypes are learned for the clustering of time-series signals. Their prototypes, however, cannot cluster instances based on multiple patient attributes and do not extend to the retrieval setting.

**Clinical information retrieval (IR)**   Retrieving clinical data from a large database has been a longstanding goal of researchers within healthcare [25]. Such research has involved the retrieval of clinical documents [26, 7, 27, 28] where, for example, Avolio *et al.* [29] map text queries to an ontology known as SNOMED, before retrieving relevant clinical documents. Recent research has focused on the retrieval of biomedical images [6, 30], and EHR data [31, 32] to discover patient cohorts in a clinical database [8]. Goodwin *et al.* [9] implement an unsupervised patient cohort retrieval system by exploiting clinical text and time-series data. These approaches, however, do not explore cardiac signals, cannot account for multiple patient attributes, and are unable to also cluster instances. To the best of our knowledge, we are the first to design a learning framework that allows for *both* the clustering and retrieval of cardiac signals based on multiple patient attributes.

## 3 Background

**Supervised clustering**   We learn a function, $f_\theta : \boldsymbol{x} \in \mathbb{R}^D \to \boldsymbol{v} \in \mathbb{R}^E$, parameterized by $\boldsymbol{\theta}$, that maps a $D$-dimensional input, $\boldsymbol{x}$, to an $E$-dimensional representation, $\boldsymbol{v}$. We also have a labelled dataset, $\mathcal{D}_l = \{\boldsymbol{x_i}, A_i\}_{i=1}^{N_l}$, where each instance, $\boldsymbol{x_i}$, is associated with a set of discrete patient attributes, $A_i = \{\alpha_c^i, \alpha_s^i, \alpha_a^i\}$ where $\alpha_c$ = disease class, $\alpha_s$ = sex and $\alpha_a$ = age. Supervised clustering can involve learning $M \ll N_l$ centroids with each representing a unique set of attributes, $\{\alpha_c^j, \alpha_s^j, \alpha_a^j\} \in \{A_j\}_{j=1}^M$, and grouping similar instances together. Given unlabelled instances, $\{\boldsymbol{x_u}\}_{u=1}^N$, the centroid closest to each representation, $\boldsymbol{v_u} = f_\theta(\boldsymbol{x_u})$, is used to infer the latter's attributes. In this work, we learn cluster centroids which are more formally introduced in Sec. 4.1.

**Information retrieval**   IR involves searching through a large, unlabelled dataset, $\{\boldsymbol{x_u}\}_{u=1}^N$, and retrieving a relevant instance, $\boldsymbol{x_u}$. However, relevance, defined based on whether an instance satisfies some criteria, is difficult to ascertain when instances are *unlabelled*. Typically, a query in the form of an embedding which represents a desired set of attributes, $A_j$, retrieves its closest (and most relevant) representation, $\boldsymbol{v_u} = f_\theta(\boldsymbol{x_u})$, and infers the latter's attributes. In this work, we learn a set of query embeddings. As will become apparent in Sec. 5, these embeddings can also be treated as centroids, like those outlined in supervised clustering, and will thus serve a dual purpose.

# 4 Methods

## 4.1 Attribute-specific clinical prototypes

We aim to learn embeddings, referred to as clinical prototypes, that can be exploited for *both* the clustering and retrieval of cardiac signals based on multiple patient attributes. In the clustering setting, the goal is to annotate *unlabelled* instances with a set of patient attributes. To that end, we exploit clinical prototypes as centroids of clusters to which such unlabelled instances are assigned (see Fig. 1 left). In the retrieval setting, the goal is to retrieve *unlabelled* instances based on a set of patient attributes. To that end, we exploit each clinical prototype as a query to search through an *unlabelled* database and retrieve instances to which it is most similar (see Fig. 1 right).

In designing clinical prototypes, we take inspiration from the field of natural language processing (NLP) where a learnable word embedding represents a unique word. In our case, each prototype represents a unique combination of discrete patient attributes. Formally, given the attributes, $\alpha_c$, $\alpha_s$, and $\alpha_a$, we would have $M = |\alpha_c| \times |\alpha_s| \times |\alpha_a|$ such unique combinations denoted by $A = \{\alpha_c^j, \alpha_s^j, \alpha_a^j\}_{j=1}^M$. We associate each combination, $A_j \in A$, with a learnable prototype, $\boldsymbol{p}_{\boldsymbol{A}_j} \in \mathbb{R}^E$, for a set of $M$ prototypes, $P = \{\boldsymbol{p}_{\boldsymbol{A}_j}\}_{j=1}^M$. Note that this framework extends to any number of attributes. In the next section, we outline how to learn these clinical prototypes.

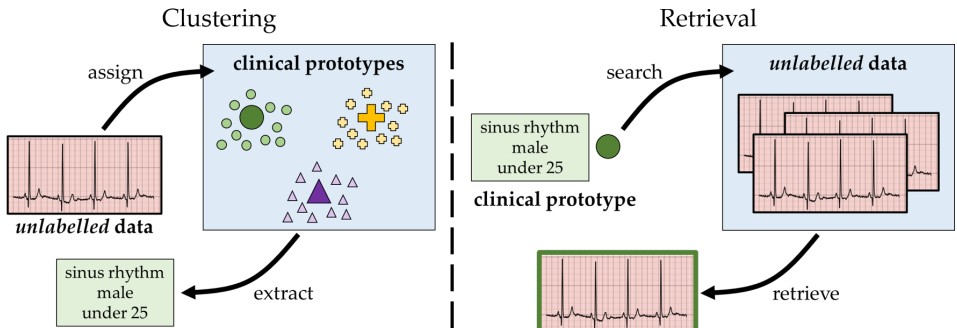

Figure 1: **Clinical prototypes are exploited for attribute-specific clustering and retrieval of cardiac signals.** For clustering, we exploit prototypes as centroids of clusters to which *unlabelled* instances are assigned. Such an assignment is associated with a set of attributes, such as disease class, sex, and age. For retrieval, we exploit each prototype as a query, associated with a set of attributes, to search through an *unlabelled* database and retrieve instances to which it is most similar.

## 4.2 Learning attribute-specific clinical prototypes

Clinical prototypes will serve a dual purpose of attribute-specific clustering and retrieval. As such, prototypes will need to be in proximity to a subset of representations of instances associated with a specific set of patient attributes. To achieve this proximity, we leverage the contrastive learning framework which involves a sequence of attractions and repulsions, as explained next.

**Hard assignment** We encourage the representation, $\boldsymbol{v_i} = f_\theta(\boldsymbol{x_i})$, of an instance, $\boldsymbol{x_i}$, associated with a set of attributes, $A_i \in A$, to be similar to the *single* clinical prototype, $\boldsymbol{p}_{\boldsymbol{A}_j}$, that shares the exact same set of attributes (i.e., $A_i = A_j$), and dissimilar to the remaining clinical prototypes, $\{\boldsymbol{p}_{\boldsymbol{A}_k}\}_{k \neq j}$. To achieve this, we optimize $\mathcal{L}_{NCE-hard}$ for a mini-batch of $B$ instances (Eq. 1). Intuitively, it heavily penalizes the learner if less probability mass is placed on the similarity of $\boldsymbol{v_i}$ and $\boldsymbol{p}_{\boldsymbol{A}_j}$ than on the similarity of other representation-and-prototype pairs. We choose to quantify the cosine similarity of pairs, $s(\boldsymbol{v_i}, \boldsymbol{p}_{\boldsymbol{A}_j})$, alongside a temperature parameter, $\tau_s$, as is done by Kiyasseh *et al.* [20].

$$\mathcal{L}_{NCE-hard} = -\frac{1}{B} \sum_{i=1}^B \log \left( \frac{e^{s(\boldsymbol{v_i}, \boldsymbol{p}_{\boldsymbol{A}_j})}}{\sum_k^M e^{s(\boldsymbol{v_i}, \boldsymbol{p}_{\boldsymbol{A}_k})}} \right) \qquad s(\boldsymbol{v_i}, \boldsymbol{p}_{\boldsymbol{A}_j}) = \frac{\boldsymbol{v_i} \cdot \boldsymbol{p}_{\boldsymbol{A}_j}}{\|\boldsymbol{v_i}\| \|\boldsymbol{p}_{\boldsymbol{A}_j}\|} \cdot \frac{1}{\tau_s} \qquad (1)$$

We refer to this many-to-one mapping from representations to clinical prototype as a hard assignment. Such an assignment, however, implies that a prototype is unlikely to extract potentially useful

information from a representation whose attributes are not *exactly* the same as those of the prototype. We quantify this limitation in Sec. 6.4 and propose an alternative assignment next.

**Soft assignment**  To overcome the limitation of a hard assignment, we encourage the representation, $v_i$, to be similar to a *subset* of clinical prototypes, $L \subset P$ (see Fig. 2). We must take caution, however, to avoid erroneously attracting representations to clinical prototypes from a *different* class. Doing so would reduce class-specific margins and thus hinder the downstream clustering and retrieval tasks.

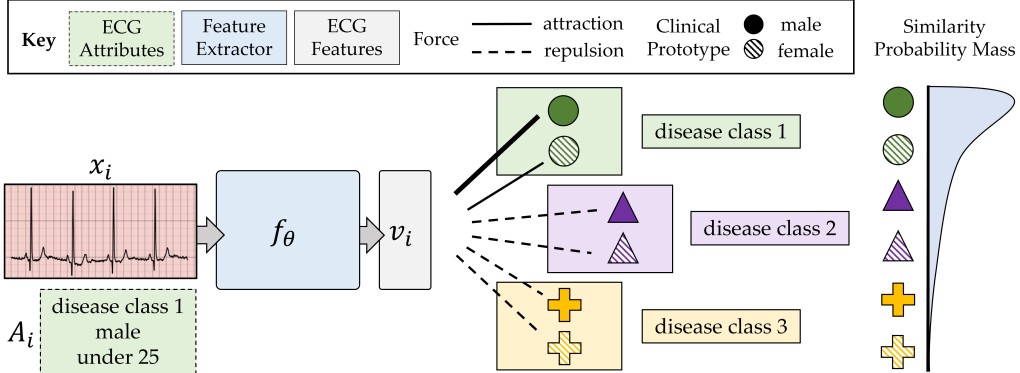

Figure 2: **Clinical prototypes are learned via a supervised contrastive learning framework referred to as CROCS.** The representation, $v_i$, of an instance, $x_i$, associated with a set of attributes, $A_i$, is strongly attracted to the clinical prototype which represents the same attributes, weakly attracted to others within the same disease class (colour), and repelled from those representing different classes. These attractions result in the shown similarity probability mass function. To avoid clutter, we have omitted the age attribute associated with clinical prototypes.

*Uniform attraction.* We chose the subset to include prototypes, $L = \{p_{A_l}\}_{l=1}^{|L|}$, that share the same disease class, $\alpha_c^i$, with the representation, $v_i$, implying that $A_l \in \{\alpha_c^i, \alpha_s^l, \alpha_a^l\}_{l=1}^{|L|} \subset A$. Note that the clinical prototypes in the subset, $L$, continue to represent attributes that vary along the dimensions of patient sex and age ($\alpha_s, \alpha_a$). Therefore, attracting $v_i$ to prototypes in $L$ *uniformly* will likely cause the latter to become minimally distinguishable across sex and age. This is an undesired outcome in light of our goal of learning *attribute-specific* prototypes. We avoid this behaviour by modulating the degree of attraction between $v_i$ and all prototypes in the set, $P$, as outlined next.

*Modulated attraction.* The attractive force between $v_i$ and $p_{A_j}$ is reflected by the corresponding $\mathcal{L}_{NCE-hard}$ term (Eq. 1). By placing less probability mass on $s(v_i, p_{A_j})$ (i.e., less similarity) than on $s(v_i, p_{A_k}) \forall k, k \neq j$, the learner incurs a higher loss and thus attracts the pair. We extend this logic to all prototypes to obtain $M$ $\mathcal{L}_{NCE-hard}$ terms per representation. To modulate these attractions, we introduce a weight, $w_{ij} \in \{w_{ij}\}_{j=1}^M$, as a coefficient of the $j$-th loss term (Eq. 2). Each weight, $w_{ij}$, quantifies the degree of matching between attributes of the representation, $A_i = \{\alpha_c^i, \alpha_s^i, \alpha_a^i\}$, and those of the clinical prototype, $A_j = \{\alpha_c^j, \alpha_s^j, \alpha_a^j\}$, as reflected by $q(A_i, A_j) \in \mathbb{R}$. We define $\mathbb{1}$ as the indicator function and $\tau_\omega$ as a temperature parameter that determines how soft the representation-and-prototype attraction is. For example, as $\tau_\omega \to \infty$, this approach reverts to the uniform attraction setup. The intuition is that a stronger attraction ($\uparrow w_{ij}$) should exist for a representation-and-prototype pair that shares more attributes. We also avoid the erroneous attraction of pairs from different classes (i.e., $\alpha_c^i \neq \alpha_c^j$) by setting $\omega_{ij} = 0$. When visualizing the UMAP projection [33] of prototypes learned with a uniform attraction ($\tau_\omega = \infty$) (Fig. 3 left) and those learned with a modulated attraction ($\tau_\omega \neq \infty$) (Fig. 3 centre), we show that the latter become more linearly separable across sex.

$$\mathcal{L}_{NCE-soft} = -\frac{1}{B}\sum_{i=1}^{B}\left[\sum_{j=1}^{M}\omega_{ij}\log\left(\frac{e^{s(v_i, p_{A_j})}}{\sum_k^M e^{s(v_i, p_{A_k})}}\right)\right] \qquad (2)$$

$$\omega_{ij} = \begin{cases} \frac{e^{q(A_i, A_j)}}{\sum_l^{|L|} e^{q(A_i, A_j)}} & \text{if } \alpha_c^i = \alpha_c^j \\ 0 & \text{otherwise} \end{cases}$$

$$q(A_i, A_j) = \frac{1}{\tau_\omega} . [\mathbb{1}(\alpha_c^i = \alpha_c^j) + \mathbb{1}(\alpha_s^i = \alpha_s^j) + \mathbb{1}(\alpha_a^i = \alpha_a^j)]$$

**Arrangement of clinical prototypes** Clinical prototypes would confer a high degree of interpretability if they also captured the semantic relationships between attributes. Concretely, prototypes representing similar attribute sets (e.g., adjacent age groups) should be similar to one another. This is analogous to the high similarity of word embeddings representing semantically similar words [34]. To capture these semantic relationships, we encourage class-specific prototypes to maintain some desired distance between one another. As such, each pair of $M$ clinical prototypes, $\boldsymbol{p_{A_j}}$, $\boldsymbol{p_{A_k}}$ $\forall j, k \in [1, M]$ is associated with an empirical and ground-truth (desired) distance. For the former, we normalize the prototypes ($L_2$ norm) and calculate their Euclidean distance, $\hat{d}_{jk} = \left\| \boldsymbol{p_{A_j}} - \boldsymbol{p_{A_k}} \right\|_2 \forall j, k \in [1, M]$. For the latter, we define the ground-truth distance as $d_{jk} = \beta \times d_H \in \mathbb{R}$, where $d_H(A_j, A_k) \in \mathbb{Z}^+$ is the Hamming distance between a pair of discrete attribute sets. Intuitively, the Hamming distance counts the number of attribute mismatches and $\beta \in \mathbb{R}$ penalizes each mismatch. Therefore, we can generate an *empirical* set, $\{\hat{d}_{jk}\}_{j,k=1}^M$ and a *ground-truth* set, $\{d_{jk}\}_{j,k=1}^M$, of distance values. By minimizing the mean-squared error between these two sets, we learn clinical prototypes that adopt a semantically meaningful arrangement (see Fig. 3 right). Since we are only interested in adopting this arrangement for prototypes of the same class (i.e., $\alpha_c^j = \alpha_c^k$), we incorporate the regularization term, $\mathcal{L}_{reg}$, into the final objective function, $\mathcal{L}_{tot}$.

$$\mathcal{L}_{reg} = \sum_{j,k=1}^M (\hat{d}_{jk} - d_{jk})^2 \Leftrightarrow \alpha_c^j = \alpha_c^k \qquad \mathcal{L}_{tot} = \mathcal{L}_{NCE-soft} + \mathcal{L}_{reg} \qquad (3)$$

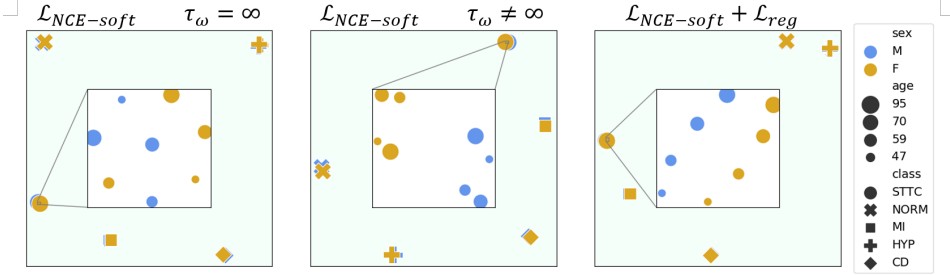

Figure 3: **Projection of the set of clinical prototypes, $P$, learned with variants of the CROCS framework.** These prototypes are derived through empirical experiments and projected onto a lower-dimensional space via UMAP [33]. The legend indicates the attributes of sex (colour), age (size of marker), and disease class (marker symbol). With $\mathcal{L}_{NCE-soft}$ $\tau_\omega = \infty$, representations are attracted to class-specific prototypes *uniformly*. With $\mathcal{L}_{NCE-soft}$ $\tau_\omega \neq \infty$, prototypes become more linearly separable across sex. Our full framework, $\mathcal{L}_{NCE-soft} + \mathcal{L}_{reg}$, leads to prototypes that adopt a semantically meaningful arrangement. We investigate the marginal impact of these design choices on performance in Sec. 6.4.

# 5 Experimental design

**Datasets** We use 1) **Chapman** [35] which consists of 12-lead ECG recordings from 10,646 patients alongside cardiac arrhythmia (disease) labels which we group into 4 major classes. 2) **PTB-XL** [36] consists of 12-lead ECG recordings from 18,885 patients alongside disease labels which we group into 5 major classes [37]. Each dataset contains patient sex and age information and is split, at the patient level, into training, validation, and test sets using a $60 : 20 : 20$ configuration. Each time-series recording is split into non-overlapping segments of 2500 samples ($\approx 5s$ in duration), as this is common for in-hospital recordings. Further details are provided in Appendix B.

**Description of clustering setting** During inference, we treat the clinical prototypes, $\{\boldsymbol{p_{A_j}}\}_{j=1}^M$, as a set of cluster centroids. We calculate the Euclidean distance between the $i$-th representation and each of the $M$ prototypes, identify the closest prototype, $\boldsymbol{p_{A_k}}$, and assign the representation to $A_k$ which we now denote by $\hat{A}_i = \{\hat{\alpha}_c^i, \hat{\alpha}_s^i, \hat{\alpha}_a^i\}$. Repeating this process for $N$ unseen instances results in

a set of assigned attribute values, $\vec{\hat{\alpha}} = \{\hat{\alpha}^i\}_{i=1}^N$, for a particular attribute, $\hat{\alpha} \in A$ (e.g., disease class). Such unseen instances would typically be *unlabelled*. For evaluation, however, we assume access to the ground-truth attribute values, $\vec{\alpha} = \{\alpha^i\}_{i=1}^N$, with which we calculate the accuracy, $\text{Acc}(\hat{\alpha})$, and the adjusted mutual information, $\text{AMI}(\hat{\alpha}) \in [0, 1]$, between $\vec{\hat{\alpha}}$ and $\vec{\alpha}$.

$$\text{Acc}(\hat{\alpha}) = \frac{1}{N} \sum_{i=1}^N \mathbb{1}(\hat{\alpha}^i = \alpha^i) \qquad \text{AMI}(\hat{\alpha}) = \frac{\left[\mathbb{MI}(\vec{\alpha}, \vec{\hat{\alpha}}) - \mathbb{E}(\mathbb{MI}(\vec{\alpha}, \vec{\hat{\alpha}}))\right]}{\mathbb{E}(\mathbb{H}(\vec{\alpha}), \mathbb{H}(\vec{\hat{\alpha}})) - \mathbb{E}(\mathbb{MI}(\vec{\alpha}, \vec{\hat{\alpha}}))} \qquad (4)$$

where $\mathbb{MI}(\vec{\alpha}, \vec{\hat{\alpha}})$ denotes the mutual information between the ground-truth and assigned set of attribute values, and $\mathbb{H}(\vec{\alpha})$ denotes the entropy of the attribute values.

**Description of retrieval setting**   During inference, we treat the clinical prototypes, $\{\boldsymbol{p_{A_j}}\}_{j=1}^M$, as a query set. We calculate the Euclidean distance between the $j$-th clinical prototype and representations of $N$ unseen instances, retrieve the $K$ closest instances, and then assign them to $A_j = \{\alpha_c^j, \alpha_s^j, \alpha_a^j\}$. Note that such instances would typically be *unlabelled*, thus precluding a simple SQL search. For evaluation, however, we assume access to the ground-truth attributes, $\{\alpha_c^i, \alpha_s^i, \alpha_a^i\}_{i=1}^K$, with which we calculate a variant of the precision at $K$ metric (Eq. 5). It checks whether at least one of the retrieved instances is relevant, where relevance is based on a partial or exact match of query and instance attributes (# attribute matches). This value is then averaged across all $M$ prototypes.

$$\text{P@}K = \frac{1}{M} \sum_{j=1}^M \mathbb{1}\left(\sum_{i=1}^K \mathbb{1}\underbrace{\left(\left[\alpha_c^i = \alpha_c^j\right] \cap \left[\alpha_s^i = \alpha_s^j\right] \cap \left[\alpha_a^i = \alpha_a^j\right]\right)}_{\text{relevance} \equiv \text{\# attribute matches} = 3} \geq 1\right) \qquad (5)$$

Ideally, a retrieved instance would match all of the query's attributes (disease class, sex, and age). In our context, however, the motivation for breaking down the evaluation of the retrieval setting based on the number of attribute matches is twofold. First, it allows us to evaluate our framework at a more granular level and thus detect subtle changes in performance. Second, the importance of such an evaluation at the attribute level arguably depends on the clinical context. For example, a cardiologist diagnosing a disease might be most interested in the pathology (i.e., disease class) attribute. On the other hand, a pharmaceutical company looking to recruit patients within a particular age group for a clinical trial might be most interested in the age attribute. Moreover, biomedical researchers looking to stratify treatment outcomes based on sex would be most interested in the sex attribute.

**Baseline methods**   We compare clinical prototypes learned via the CROCS framework (**CP CROCS**) to the following methods. For retrieval, **Deep Transfer Cluster (DTC)** [38] learns cluster prototypes by minimizing the KL divergence between a target distribution and one based on the distance between prototypes and representations. **TP CROCS** involves traditional prototypes where each prototype, $\bar{\boldsymbol{v}}_{\boldsymbol{A_j}} = \frac{1}{\sum \mathbb{I}(A_i = A_j)}) \sum_{i=1}^N \boldsymbol{v_i} \cdot \mathbb{I}(A_i = A_j)$, is simply an average of representations, $\boldsymbol{v_i}$, associated with the same set of attributes, $A_j$. Such representations are also learned via CROCS.

For the clustering task, we compare to several state-of-the-art clustering methods, in addition to those mentioned above. $k$-means identifies cluster centroids based on the input instances, $\boldsymbol{x}$ (**KM raw**), or representations, $\boldsymbol{v}$, learned via the CROCS (**KM CROCS**) or explainable prototypes (**KM EP**) [11] framework. **DeepCluster (DC)** [39] iteratively applies $k$-means to representations, pseudo-labels them according to their assigned cluster, and then exploits such labels for supervised training. **Deep Temporal Clustering Representation (DTCR)** [16] optimizes an objective function with a reconstruction, $k$-means, and classifier loss that determines whether instances are real. **Information Invariant Clustering (IIC)** [40] maximizes the mutual information between class probabilities of an instance and its perturbed counterpart. **SeLA** [41] implements Sinkhorn-Knopp to pseudo-label instances in a supervised manner. For further details, see Appendix D.

**Hyperparameters**   During optimization, we chose the temperature parameter, $\tau_s = 0.1$ [20], and $\tau_w = 1$. We specified $\beta = 0.2$, in the regularization term, after experimenting with several values (see Appendix F). Too small a value of $\beta$ would decrease the distance between class-specific clinical prototypes. Too large a value of $\beta$ would cause clinical prototypes from *different* classes to overlap with one another and thus reduce class separability. For Chapman and PTB-XL, sex $\in \{\text{M}, \text{F}\}$, age is converted to quartiles, and $|\text{class}| = 4$ and 5, respectively. Therefore, $M = |\text{class}| \times |\text{sex}| \times |\text{age}| =$

32 and 40, for the two datasets, respectively. The network, $f_\theta$, comprises 1D convolutional operators and we chose the embedding dimension $E = 128$ and 256 for Chapman and PTB-XL, respectively. We show that $E$ has a minimal effect on performance (see Appendix E). Further network and implementation details can be found in Appendix C.

## 6 Experimental results

### 6.1 Visualizing clinical prototypes

We begin by qualitatively validating the claim that clinical prototypes are attribute-specific. In other words, can prototypes be delineated along the dimensions of disease class, sex, and age? To address this, we illustrate, in Fig. 4, two dimensional UMAP projections of the class-specific clinical prototypes (large, coloured shapes), traditional prototypes (large, black shapes), and representations of instances in the validation set of Chapman and PTB-XL.

We show that clinical prototypes are indeed disease class-specific, as evident by the high degree of class separability of such prototypes. We also find that clinical prototypes are distinct from traditional prototypes, a distinction whose importance will become evident in later sections. Second, the consistency of the class labels of the prototypes with those of the representations is a harbinger of how prototypes may perform in the clustering and retrieval settings, as we show in the next section. These findings complement the delineation of the prototypes along the dimensions of sex and age, and their adoption of a semantically meaningful arrangement, as was shown in Section 4.

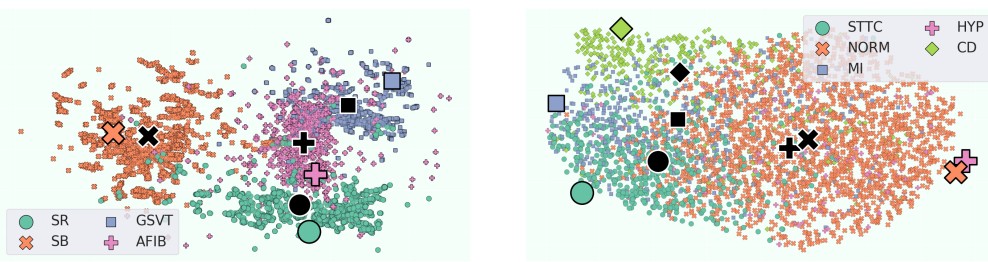

Figure 4: **Projection of class-specific clinical prototypes $p$ (large, coloured shapes), traditional prototypes $\bar{v}$, (large, black shapes), and representations, $v$, in the validation set of (left) Chapman and (right) PTB-XL.** We show that clinical prototypes are class-specific, consistent with the class labels of representations, and distinct from traditional prototypes. This bodes well for their use as centroids for clustering (see Sec. 6.2) and as queries for retrieval (see Sec. 6.3).

### 6.2 Deploying clinical prototypes in clustering setting

In the clustering setting, we assign cardiac signals in a held-out dataset to a set of patient attributes associated with the cluster of the closest clinical prototype. We evaluate these assignments based on the three patient-specific attributes (disease class, sex, and age) and present the results in Table 1.

In Table 1, we find that CROCS outperforms both generic and domain-specific state-of-the-art clustering methods. For example, on Chapman, CP CROCS, KM EP, and DTC achieve Acc(class) = 90.3, 65.6, and 53.4% respectively. Along the dimension of sex, and on PTB-XL, CP CROCS and DTC achieve Acc(sex) = 73.5 and 58.6%, respectively. Second, we find that CROCS leads to rich representation learning that facilitates clustering. This is evident when comparing the performance of $k$-means applied to representations that are learned via different methods. For example, on Chapman, KM raw, KM EP, and KM CROCS achieve Acc(class) = 28.4, 65.6, and 73.4%, respectively. We also find that clinical prototypes, when exploited as centroids, are preferable to traditional prototypes, and centroids learned via $k$-means. For example, on PTB-XL, KM CROCS, TP CROCS, and CP CROCS achieve Acc(class) = 47.6, 53.6, and 76.0%, respectively. These findings, which hold across datasets and evaluation metrics, point to the overall utility of the CROCS framework and clinical prototypes for attribute-specific clustering.

Table 1: **Clustering performance on the validation set of Chapman and PTB-XL.** Evaluation is based on (a) class and (b) sex and age attributes. Results are averaged across five random seeds. Brackets indicate standard deviation and bold reflects the top-performing method. We show that CP CROCS outperforms the remaining methods regardless of patient attribute.

(a) Cardiac arrhythmia class attribute

| Method | Chapman | | PTB-XL | |
|---|---|---|---|---|
| | Acc | AMI | Acc | AMI |
| SeLA [41] | 21.0 (0.1) | 9.2 (10.0) | 10.5 (0.1) | 1.6 (0.5) |
| DC [39] | 21.0 (0.1) | 3.0 (4.0) | 10.5 (0.1) | 4.9 (0.0) |
| IIC [40] | 27.0 (0.2) | 0.2 (0.0) | 22.0 (0.7) | 0.5 (0.0) |
| DTCR [16] | 29.3 (1.3) | 0.2 (0.2) | 38.4 (4.2) | 1.4 (0.3) |
| DTC [38] | 53.4 (15.0) | 23.4 (20.0) | 67.3 (1.0) | 25.2 (0.7) |
| KM raw | 28.4 (1.2) | 0.3 (0.0) | - | - |
| KM EP [11] | 65.6 (4.0) | 42.8 (2.6) | 48.9 (1.1) | 24.2 (1.9) |
| KM CROCS | 73.4 (7.1) | 58.6 (2.8) | 47.6 (3.9) | 25.9 (1.1) |
| TP CROCS | 80.3 (1.4) | 65.0 (0.6) | 53.6 (0.7) | 29.1 (0.2) |
| CP CROCS | **90.3** (0.8) | **72.8** (1.6) | **76.0** (0.3) | **35.9** (0.4) |

(b) Sex and age attributes

| Method | Chapman | | PTB-XL | |
|---|---|---|---|---|
| | sex | age | sex | age |
| DTCR [16] | 51.2 (0.9) | 25.9 (0.1) | 51.0 (0.9) | 25.1 (0.8) |
| DTC [38] | 54.8 (0.5) | 26.4 (0.8) | 58.6 (1.9) | 25.7 (0.7) |
| KM EP [11] | 56.1 (0.0) | 31.0 (0.4) | 54.1 (0.3) | 29.2 (2.0) |
| KM CROCS | 54.9 (0.8) | 32.3 (0.5) | 51.8 (1.2) | 31.6 (1.5) |
| TP CROCS | 54.8 (1.0) | 31.1 (1.7) | 69.7 (0.8) | **39.4** (0.4) |
| CP CROCS | **57.4** (1.2) | **38.0** (0.8) | **73.5** (0.6) | 19.5 (0.2) |

## 6.3 Deploying clinical prototypes in the retrieval setting

Up until now, we have shown that CROCS leads to accurate clustering. In this section, we show that CROCS can also be independently exploited for retrieval. Specifically, a query retrieves the closest $K = [1, 5, 10]$ previously unseen cardiac signals, and assigns them to its associated set of patient attributes. In Table 2, we evaluate these assignments based on both partial and exact matches of the attributes (# attribute matches) represented by the query and retrieved cardiac signals.

In Table 2, we find that CROCS outperforms the baseline retrieval method, DTC. For example, on Chapman, at $K = 1$, and when # attribute matches $\geq 1$, CP CROCS and DTC achieve a precision of 95.6 and 71.9%, respectively. This indicates that, on average, 95.6% of the cardiac signals retrieved by the clinical prototypes are relevant. Relevance, in this case, implies that the retrieved cardiac signals share at least one attribute with the query. Such a finding points to the utility of clinical prototypes as queries in the retrieval setting. We also find that CROCS leads to rich representation learning that facilitates retrieval. This is evident by the strong performance of TP CROCS which depends directly on representations learned via our CROCS framework. For example, on PTB-XL, at $K = 1$, and when # attribute matches $\geq 1$, DTC, TP CROCS, and CP CROCS achieve a precision of 70.0, 99.0, and 92.5%, respectively. In this particular case, the lower performance of CP CROCS relative to TP CROCS is hypothesized to stem from clinical prototypes acting instead as *archetypes* (extreme representative data points) [42] which may occasionally hinder retrieval along multiple attributes. Evidence of such extreme embeddings can be found in Fig. 4. We also show that CROCS continues to perform well even when provided with only 10% of the labelled training data (see Appendix G).

Table 2: **Precision of $K$ retrieved representations, $v$, in the validation set of Chapman and PTB-XL, that are closest to the query.** Results are shown for partial and exact matches of the attributes (# attribute matches) represented by the query and retrieved cardiac signals, and are averaged across five random seeds. Brackets indicate standard deviation and bold reflects the top-performing method. The strong performance of CP CROCS provides evidence in support of our CROCS framework.

| # attribute matches | Query | Chapman | | | PTB-XL | | |
|---|---|---|---|---|---|---|---|
| | | $K = 1$ | 5 | 10 | 1 | 5 | 10 |
| $\geq 1$ | DTC [38] | 71.9 (0.0) | 100.0 (0.0) | 100.0 (0.0) | 70.0 (0.0) | 90.0 (8.4) | 100.0 (0.0) |
| | TP CROCS | 91.9 (3.2) | 97.5 (2.3) | 100.0 (0.0) | **99.0** (2.0) | 100.0 (0.0) | 100.0 (0.0) |
| | CP CROCS | **95.6** (6.1) | 100.0 (0.0) | 100.0 (0.0) | 92.5 (0.0) | 100.0 (0.0) | 100.0 (0.0) |
| $\geq 2$ | DTC [38] | 25.0 (0.0) | 71.9 (9.7) | 90.0 (7.0) | 22.5 (0.0) | 52.5 (16.0) | 80.5 (4.0) |
| | TP CROCS | 55.0 (3.8) | 79.4 (7.6) | 90.0 (0.1) | **71.5** (2.0) | 94.5 (1.0) | **100.0** (0.0) |
| | CP CROCS | **61.3** (10.0) | **86.3** (10.9) | **93.8** (7.9) | 63.0 (1.9) | **96.5** (2.0) | 99.5 (1.0) |
| $= 3$ | DTC [38] | 3.1 (0.0) | 15.0 (3.1) | 23.8 (0.1) | 2.5 (0.0) | 9.5 (4.3) | 16.5 (0.2) |
| | TP CROCS | 10.6 (1.5) | 23.8 (6.1) | 36.9 (7.0) | **15.5** (1.0) | 32.0 (1.0) | 43.5 (0.1) |
| | CP CROCS | **11.3** (2.5) | **33.1** (6.1) | **46.3** (6.7) | 12.5 (0.0) | **33.5** (2.0) | **43.0** (4.0) |

To qualitatively evaluate the retrieval performance, we first randomly choose a query representing a set of attributes and calculate its Euclidean distance to the representations in a validation set. We present

distributions of such distance values in Fig. 5 (top row), for a DTC query and a CP CROCS query, coloured based on the ground-truth class of the representations (other queries shown in Appendix H). In Fig. 5 (bottom row), we illustrate the six cardiac signals ($K = 6$) that are closest to each query, with a green border indicating signals whose class attribute matches that of the query. We find that the CP CROCS query is closer to representations of the same class (SR) than to those of a different class. For example, in Fig. 5b (top row), the average distance between the CP CROCS query representing {SR, male, under 49} and representations with and without the class attribute SR is $\approx 0.6$ and $> 1.5$, respectively. Such separability, which is not exhibited by the DTC query, points to the improved reliability of the CP CROCS query in distinguishing between the relevance of cardiac signals. Further evidence in support of this reliability is shown in Fig. 5 (bottom row) where we find that a DTC and a CP CROCS query retrieve relevant cardiac signals 0% and 50% of the time, respectively. This finding also extends to the PTB-XL dataset (Appendix H).

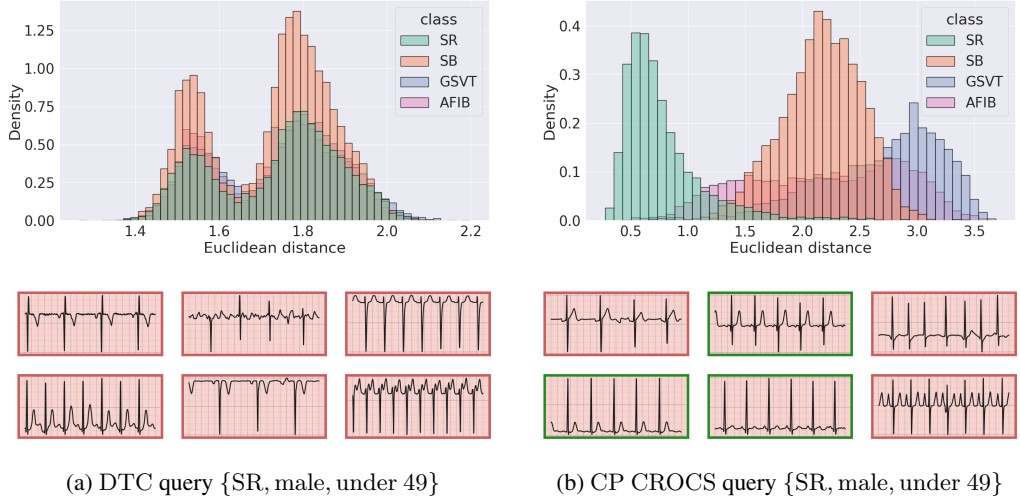

(a) DTC query {SR, male, under 49}  (b) CP CROCS query {SR, male, under 49}

Figure 5: **Qualitative retrieval performance of two distinct query prototypes.** (**top row**) Euclidean distance from (a) DTC query or (b) CP CROCS query to representations, $v$, in the validation set of Chapman. (**bottom row**) Six closest cardiac signals to the query which is associated with a set of patient attributes {disease, sex, age}. Retrieved cardiac signals with a green border indicate those whose class attribute matches that of the query. We show that the CP CROCS query is closer to representations of the same class (SR) and thus retrieves relevant cardiac signals.

## 6.4 Investigating the marginal impact of design choices

We have shown that CROCS reliably allows for both clustering and retrieval. In this section, we conduct several ablation studies to better understand the root cause of this reliability (see Table 3). We find that, on average, the soft assignment of representations to prototypes is preferable to the hard

Table 3: **Marginal impact of design choices of CROCS on clustering performance.** Evaluation is based on (a) class and (b) sex and age attributes. Results are averaged across five random seeds. Brackets indicate standard deviation and bold reflects the top-performing method. We show that our full framework ($\mathcal{L}_{NCE-soft} + \mathcal{L}_{reg}$) is preferable to other variants regardless of attribute.

(a) Cardiac arrhythmia class attribute

| Method | Chapman | | PTB-XL | |
|---|---|---|---|---|
| | Acc | AMI | Acc | AMI |
| $\mathcal{L}_{NCE-hard}$ | 86.8 (0.7) | 67.5 (1.1) | 66.5 (0.1) | 35.0 (0.0) |
| $\mathcal{L}_{NCE-soft}$ | | | | |
| $\tau_\omega = \infty$ | 87.3 (0.5) | 68.2 (0.6) | 76.3 (0.5) | 36.1 (1.0) |
| $\tau_\omega \neq \infty$ | 89.8 (1.7) | 72.1 (2.8) | 76.1 (0.2) | 36.0 (0.4) |
| $+\mathcal{L}_{reg}$ | **90.3** (0.8) | **72.8** (1.6) | 76.0 (0.3) | 35.9 (0.4) |

(b) Sex and age attributes

| Method | Chapman | | PTB-XL | |
|---|---|---|---|---|
| | sex | age | sex | age |
| $\mathcal{L}_{NCE-hard}$ | 56.9 (0.2) | 26.2 (0.0) | 76.3 (0.7) | 19.8 (0.0) |
| $\mathcal{L}_{NCE-soft}$ | | | | |
| $\tau_\omega = \infty$ | 55.2 (0.5) | 34.7 (0.3) | 50.4 (0.1) | 20.8 (1.0) |
| $\tau_\omega \neq \infty$ | 56.8 (1.8) | 37.4 (1.0) | 74.3 (0.0) | 19.2 (0.9) |
| $+\mathcal{L}_{reg}$ | **57.4** (1.2) | **38.0** (0.8) | 73.5 (0.6) | 19.5 (0.2) |

assignment. For example, on PTB-XL, $\mathcal{L}_{NCE-soft}$ and $\mathcal{L}_{NCE-hard}$ achieve Acc(class) $\approx 76.0$ and 66.5%, respectively. We also find that our full framework ($\mathcal{L}_{NCE-soft} + \mathcal{L}_{reg}$) performs better than, or on par with, other variants. For example, on Chapman, $\mathcal{L}_{NCE-hard}$ and $\mathcal{L}_{NCE-soft}$ $\tau_\omega = \infty$, and $\tau_\omega \neq \infty$ achieve AMI(class) = 67.5, 68.2, and 72.1%, respectively, whereas $\mathcal{L}_{NCE-soft} + \mathcal{L}_{reg}$ achieves AMI(class) = 72.8%. This is a positive outcome given that the regularization term's main purpose was simply to improve the interpretability of prototypes by allowing them to capture the semantic relationships between attributes. These findings extend to the retrieval setting (Appendix I).

## 7 Discussion

In this paper, we proposed a supervised contrastive learning framework, entitled CROCS, for the clustering and retrieval of cardiac signals based on multiple patient attributes. In the process, we attracted representations associated with a set of attributes to learnable embeddings, termed clinical prototypes, that share such attributes and repelled them from prototypes with different attributes. We showed that CROCS outperforms the state-of-the-art method, DTC, when clustering, and retrieves relevant cardiac signals while lending itself to a higher degree of interpretability.

We acknowledge several limitations that can be addressed in future work. We made the design choice of discretizing patient attributes (e.g., sex and age). However, these attributes, and indeed other clinical parameters, can be continuous. Therefore, extending clinical prototypes to capture this continuity could offer researchers finer control over the clustering and retrieval process. In doing so, we envision two main challenges. First, continuous attribute values imply that the number of prototypes to be learned will grow. This may pose a computational bottleneck. To alleviate this bottleneck, researchers may benefit from advancements in the field of NLP and textual representation learning. Second, to learn useful prototypes in this continuous setting, one may require sufficient labelled data associated with each attribute combination. In practice, this may be difficult to achieve, particularly for under-represented attribute groups. Furthermore, in this work, we were limited to three discrete patient attributes. Such a decision, although primarily motivated by the availability of patient meta-data alongside ECG signals, does not exploit the diverse and abundant meta-data typically available within healthcare.

## Acknowledgements

We thank the anonymous reviewers and Antong Chen for their insightful feedback. We also thank Nagat Al-Saghira and Mohammed Abdel Wahab for lending us their voice. David Clifton was supported by the EPSRC under Grants EP/P009824/1and EP/N020774/1, and by the National Institute for Health Research (NIHR) Oxford Biomedical Research Centre (BRC). The views expressed are those of the authors and not necessarily those of the NHS, the NIHR or the Department of Health. Tingting Zhu was supported by the Engineering for Development Research Fellowship provided by the Royal Academy of Engineering.

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
