# B  Datasets

## B.1  Data pre-processing

For all of the datasets, frames consisted of 2500 samples and consecutive frames had no overlap with one another. Data splits were always performed at the patient-level.

**Chapman** [35]. Each ECG recording was originally 10 seconds with a sampling rate of 500Hz. We downsample the recording to 250Hz and therefore each ECG frame in our setup consisted of 2500 samples. We follow the labelling setup suggested by [35] which resulted in four classes: Atrial Fibrillation, GSVT, Sudden Bradychardia, Sinus Rhythm. The ECG frames were normalized in amplitude between the values of 0 and 1.

**PTB-XL** [36]. Each ECG recording was originally 10 seconds with a sampling rate of 500Hz. We extract 5-second non-overlapping segments of each recording generating frames of length 2500 samples. We follow the diagnostic class labelling setup suggested by [37] which resulted in five classes: Conduction Disturbance (CD), Hypertrophy (HYP), Myocardial Infarction (MI), Normal (NORM), and Ischemic ST-T Changes (STTC). Furthermore, we only consider ECG segments with one label assigned to them. The ECG frames were standardized to follow a standard Gaussian distribution.

## B.2  Data samples

In this section, we outline the number of instances used during training, validation, and testing for the Chapman and PTB-XL datasets (see Table 4).

Table 4: Number of instances (number of patients) used during training. These represent sample sizes for all 12 leads.

| Dataset | Train | Validation | Test |
|---------|-------|------------|------|
| Chapman | 76,614 (6,387) | 25,524 (2,129) | 25,558 (2,130) |
| PTB-XL | 22,670 (11,335) | 3,284 (1,642) | 3,304 (1,152) |

# C Implementation details

In this section, we outline the neural network architectures used for our experiments. More specifically, we use the architecture shown in Table 5 for all experiments pertaining to the Chapman dataset. Given the size of the PTB-XL dataset and the relative complexity of the corresponding task (at least at the disease class level), we opted for a more complex network. We modified the ResNet18 architecture whereby the number of blocks per layer was reduced from two to one, effectively reducing the number of parameters by a factor of two. We chose this architecture after experimenting with several variants. Experiments were conducted using PyTorch [43] and an NVIDIA Quadro RTX 6000 GPU. Each training and validation epoch took approximately 2 minutes, and 20 seconds to complete, respectively.

Table 5: Network architecture used for experiments conducted on the Chapman dataset. $K$, $C_{in}$, and $C_{out}$ represent the kernel size, number of input channels, and number of output channels, respectively. A stride of 3 was used for all convolutional layers. $E$ represents the dimension of the final representation.

| Layer Number | Layer Components | Kernel Dimension |
|---|---|---|
| 1 | Conv 1D BatchNorm ReLU MaxPool(2) Dropout(0.1) | 7 x 1 x 4 ($K$ x $C_{in}$ x $C_{out}$) |
| 2 | Conv 1D BatchNorm ReLU MaxPool(2) Dropout(0.1) | 7 x 4 x 16 |
| 3 | Conv 1D BatchNorm ReLU MaxPool(2) Dropout(0.1) | 7 x 16 x 32 |
| 4 | Linear ReLU | 320 x $E$ |

Table 6: Batchsize and learning rates used for training with different datasets. The Adam optimizer was used for all experiments.

| Dataset | Batchsize | Learning Rate |
|---|---|---|
| Chapman | 256 | $10^{-4}$ |
| PTB-XL | 128 | $10^{-5}$ |

# D   Baseline implementations

**DeepCluster**   In the implementation by Caron *et al.* [39], a forward pass of each instance in the training set is performed. This generates a set of representation which are then clustered, in an unsupervised manner, using $k$-means. This involves a decision regarding the value of K, i.e., the number of clusters. In our supervised setting, we have this information available and therefore set the value of K to be equal to the number of distinct cardiac arrhythmia classes. Once the clustering is complete, each instance is assigned a pseudo-label according to the cluster to which it belongs. Such pseudo-labels are used as the ground-truth for supervised training during the next epoch. We repeat this process after each epoch for a total of 30 epochs after realizing that the validation loss plateaus at that point.

**IIC**   In this implementation, the network is tasked with maximizing the mutual information between the representation of an instance and that of its perturbed counterpart. Such perturbations must be class-preserving and, in computer vision, consist of random crops, rotations, and modifications to the brightness of the images. In our setup involving time-series data, we perturb instances by using additive Gaussian noise in order to avoid erroneously flipping the class of a particular instance. In addition to the aforementioned, we implement the auxiliary over-clustering method suggested by the authors. This approach allows one to model additional 'distractor' classes that may be present in the dataset, and was shown by [40] to improve generalization performance. In our setup, we set the number of total clusters to the number of attribute combinations, $M$.

**SeLA**   In this implementation, each instance is *assigned* a posterior probability distribution. For all instances, this results in an assigned matrix of posterior probability distributions. Each instance's label is obtained by identifying the index associated with the largest posterior probability distribution. Deriving the aforementioned matrix is the crux of SeLA. It does by solving the Sinkhorn-Knopp algorithm under the assumption that the dataset can be evenly split into K clusters. Our setup does not deviate from the original implementation found in [41].

**DeepTransferCluster**   In this implementation, the distance between each representation and each cluster prototype is calculated to generate a probability distribution over classes, $p$. The distribution, $p$, is encouraged to be similar to a target distribution, $z$, by minimizing the KL divergence of these two distributions. In the original unsupervised implementation, the target distribution is a sharper version of the empirical distribution [38]. In our supervised implementation, we initialize the prototypes similarly to our approach and modify the target distribution to incorporate labels. As with our soft-assignment, we aim for a target distribution that reflects discrepancies, $d$, between the representation attributes, $A_i$, and the prototype attributes, $A_j$. Mathematically, our target distribution, $z$, is as follows:

$$z_j = \frac{e^{\omega_{ij}}}{\sum_l^{|L|} e^{\omega_{il}}} \tag{6}$$

$$\omega_{ij} = \begin{cases} \frac{e^{d(A_i, A_j)}}{\sum_l^{|L|} e^{d(A_i, A_l)}} & \text{if } \alpha_1^i = \alpha_1^j \\ 0 & \text{otherwise} \end{cases} \tag{7}$$

$$d(A_i, A_j) = \frac{1}{\tau_\omega} \cdot [\delta(\alpha_c^i = \alpha_c^j) + \delta(\alpha_s^i = \alpha_s^j) + \delta(\alpha_a^i = \alpha_a^j)] \tag{8}$$

**K-means EP**   In this implementation by Gee *et al.* [11], each instance is first passed through the encoder network to generate a representation. This representation serves multiple functions: a) it is passed through the decoder network to reconstruct the input, and b) passed through a prototype network that works as follows. The Euclidean distance between the representation and $M$ randomly-initialized embeddings (prototypes) is calculated to generate a single $M$-dimensional representation. This newly-generated representation is then passed through a linear classification head to predict the cardiac arrhythmia class associated with the original instance. In our setup, we set the number of prototypes to coincide with the number of clinical prototypes that we use. For clustering, we apply the $k$-means algorithm to the representations learned via this framework.

**Deep Temporal Clustering Representation**   In this implementation by Ma *et al.* [16], the network consists of three main components: 1) an encoder, 2) a decoder, and 3) a classifier head. A synthetic version of each instance is first generated by permuting a certain fraction, $\alpha$, of the time-points in the original instance. The original instance and its synthetic counterpart are then passed through the encoder to obtain a pair of representations (a real and synthetic one). The classifier is tasked with identifying whether such representations are real or fake (binary classification akin to discriminator in generative adversarial networks). Moreover, the decoder reconstructs the original instance by operating on the real representation. Lastly, the $k$-means loss is approximated based on the Gram matrix of the mini-batch of real representations. We follow the original implementation, and choose $\alpha = 0.2$, and $\lambda = 10^{-3}$ as the coefficient of the $k$-means loss in the objective function.

# E   Effect of embedding dimension, $E$, on clustering

In this section, we explore the effect of the embedding dimension, $E$, on the clustering performance of our framework. Specifically, we experiment with $E \in \{32, 64, 128, 256\}$ on the Chapman dataset and present the accuracy of the attribute assignments (disease class, sex, and age) in Fig. 6. We find that the embedding dimension has minimal impact on the clustering performance of our framework when evaluated on the disease class and sex patient attributes. This is evident in Fig. 6a where the $\mathrm{Acc(class)} \approx 0.85$ across all embedding dimensions and in Fig. 6b where the $\mathrm{Acc(class)} \approx 0.55$ across all embedding dimensions. We do, however, find that an embedding dimension, $E = 128$, is favourable when evaluating the clustering performance based on the patient age assignments. This can be seen in Fig. 6c where $\mathrm{Acc(class)} \approx 0.38$ at $E = 128$, whereas $\mathrm{Acc(class)} < 0.34$ for the remaining embedding dimensions.

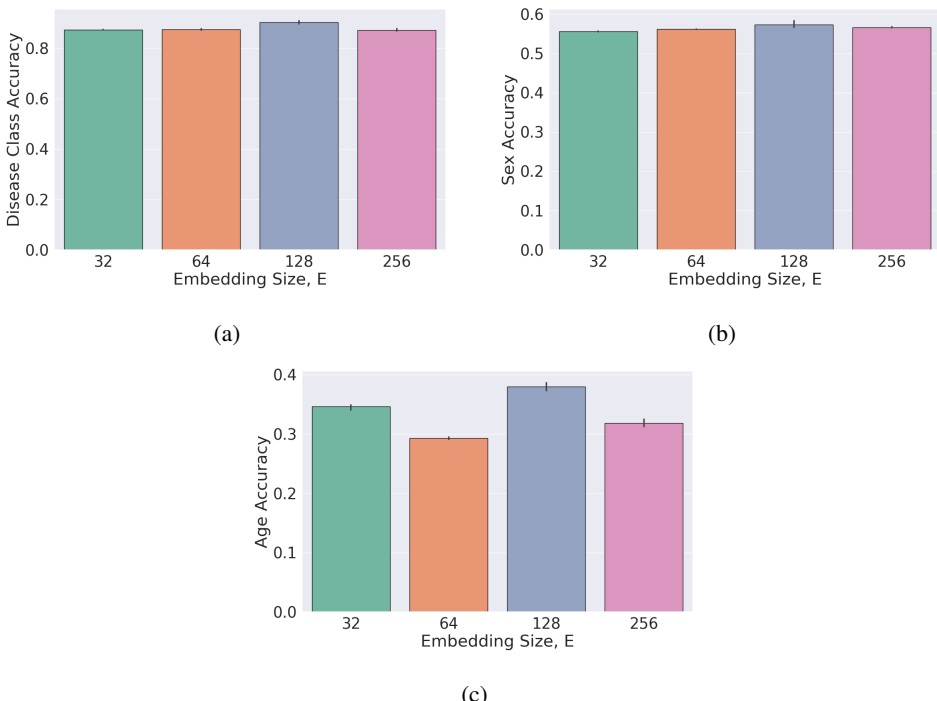

Figure 6: **Effect of embedding dimension, $E$, on the clustering performance of our framework.** Results are shown for the (a) disease class, (b) sex, and (c) age attributes across five random seeds. The error bars represent one standard deviation from the mean. We find that the embedding dimension has a minimal effect on the performance when evaluating based on disease class and sex attributes. An effect is more pronounced when clustering based on the age attribute.

## F Effect of $\beta$ on clustering and retrieval

In this section, we examine the effect of $\beta$, as used in the regularization term (Eq. 3), on the clustering and retrieval performance of our framework. We conduct the same clustering and retrieval experiments as those found in the main manuscript and experiment with $\beta = [0.05, 0.1, 0.2, 0.4]$. The results of these experiments are presented in Tables 7 and 8. In both settings, we find that $\beta = 0.2$ is preferable to the remaining values of $\beta$. This is evident by the higher clustering and retrieval performance. For example, at $\beta = 0.2$, $\text{Acc(class)} = 76.0$ whereas at the remaining $\beta$ values, $\text{Acc(class)} \approx 66.0$, reflecting a difference of $11\%$. Furthermore, in the retrieval setting, with $K = 1$ and the # attribute matches $\geq 1$, the precision at $\beta = 0.2$ is 68.8 whereas at the remaining values of $\beta$, the precision $< 67.5$. These findings are consistent with our expectations given that $\beta$ controls the distance between clinical prototypes, which, in turn, impacts their utility as centroids for clustering and as queries for retrieval.

Table 7: Effect of $\beta$ on the clustering performance on representations in the validation set of PTB-XL. Evaluation is based on (a) class attribute and (b) sex and age attributes. Results are averaged across five random seeds. Brackets indicate standard deviation and bold reflects the top-performing $\beta$ value.

(a) Cardiac arrhythmia class attribute

| $\beta$ | PTB-XL | |
| --- | --- | --- |
| | Acc | AMI |
| 0.05 | 66.0 (0.3) | 33.3 (0.0) |
| 0.1 | 65.7 (0.0) | 33.0 (0.6) |
| 0.2 | **76.0** (0.3) | **35.9** (0.4) |
| 0.4 | 65.9 (0.6) | 34.7 (0.2) |

(b) Sex and age attributes

| Method | PTB-XL | |
| --- | --- | --- |
| | Sex | Age |
| 0.05 | 72.2 (0.2) | 32.4 (0.0) |
| 0.1 | 72.9 (0.8) | 31.6 (0.5) |
| 0.2 | **73.5** (0.6) | 19.5 (0.2) |
| 0.4 | 72.5 (0.8) | 32.8 (0.2) |

Table 8: Effect of $\beta$ on precision of CROCS when retrieving the closest $K$ representations from the validation set of PTB-XL. Results shown are based on the number of attributes shared between the prototypes and the retrieved cardiac signals, and are averaged across five random seeds. Brackets indicate standard deviation and bold reflects the top-performing $\beta$ value.

| # attribute matches | $\beta$ | PTB-XL | | |
| --- | --- | --- | --- | --- |
| | | $K = 1$ | 5 | 10 |
| $\geq 1$ | 0.05 | 63.5 (0.0) | 100.0 (0.0) | 100.0 (0.0) |
| | 0.1 | 63.5 (7.0) | 100.0 (0.0) | 100.0 (0.0) |
| | 0.2 | **68.8** (8.1) | 100.0 (0.0) | 100.0 (0.0) |
| | 0.4 | 67.5 (7.7) | 97.0 (2.4) | 100.0 (0.0) |
| $\geq 2$ | 0.05 | 18.0 (4.5) | 55.0 (5.0) | 69.0 (9.3) |
| | 0.1 | 15.5 (1.0) | 49.5 (4.0) | 70.5 (6.2) |
| | 0.2 | **19.4** (6.4) | **78.8** (11.9) | **93.8** (4.4) |
| | 0.4 | 13.0 (4.0) | 50.0 (1.6) | 80.5 (1.0) |
| $= 3$ | 0.05 | 1.5 (2.0) | 6.0 (2.0) | 10.0 (0.0) |
| | 0.1 | 0.5 (1.0) | 5.5 (1.0) | 8.0 (1.0) |
| | 0.2 | 1.3 (2.5) | **14.4** (7.0) | **26.3** (8.0) |
| | 0.4 | 0.5 (1.0) | 2.5 (0.0) | 10.0 (1.6) |

# G   Performance of CROCS with Less Labelled Data

In this section, we explore the effect of less labelled data on the performance of CROCS. More precisely, we reduce the amount of labelled training data 10-fold while keeping the amount of unlabelled data fixed. In Tables 9 and 10, we present the results of these experiments in the clustering and retrieval settings, respectively.

Table 9: **Clustering performance on the validation set of Chapman and PTB-XL when CROCS is trained with 10% of labelled data.** Evaluation is based on (a) class and (b) sex and age attributes. Results are averaged across five random seeds. Brackets indicate standard deviation and bold reflects the top-performing method. The asterisk ($*$) indicates that the algorithm could not be solved. We show that CP CROCS outperforms the remaining methods regardless of patient attribute.

(a) Cardiac arrhythmia class attribute

| Method | Chapman | | PTB-XL | |
| --- | --- | --- | --- | --- |
| | Acc | AMI | Acc | AMI |
| SeLA [41] | $*$ | $*$ | 0.10 (0.0) | 0.02 (0.0) |
| DC [39] | 21.0 (0.0) | 0.5 (0.6) | 10.5 (0.0) | 4.3 (0.9) |
| IIC [40] | 27.2 (0.3) | 0.6 (0.01) | 24.3 (2.8) | 0.4 (0.7) |
| DTCR [16] | 34.3 (0.9) | 3.3 (0.0) | 24.1 (0.5) | 0.8 (0.3) |
| DTC [38] | 46.3 (2.6) | 11.8 (2.2) | 48.4 (3.9) | 0.3 (0.5) |
| KM raw | 28.4 (1.2) | 0.3 (0.0) | - | - |
| KM EP [11] | 64.9 (4.7) | 44.3 (3.4) | 45.1 (1.9) | 20.7 (1.3) |
| KM CROCS | 71.1 (4.6) | 52.7 (2.3) | 47.7 (3.4) | 20.1 (0.9) |
| TP CROCS | 75.5 (0.2) | 56.8 (0.5) | 47.3 (2.8) | 19.9 (1.4) |
| CP CROCS | **82.7** (0.4) | **61.8** (0.8) | **71.4** (0.0) | **28.8** (0.8) |

(b) Sex and age attributes

| Method | Chapman | | PTB-XL | |
| --- | --- | --- | --- | --- |
| | sex | age | sex | age |
| DTCR [16] | 52.2 (1.2) | 26.7 (0.3) | 51.7 (1.4) | 29.3 (1.2) |
| DTC [38] | 53.1 (0.6) | 26.8 (0.0) | 52.2 (0.0) | 33.5 (1.6) |
| KM EP [11] | 56.4 (0.1) | 30.2 (0.9) | 51.0 (0.7) | 37.1 (1.5) |
| KM CROCS | **56.2** (0.1) | **30.9** (0.8) | 51.1 (0.7) | 36.0 (1.1) |
| TP CROCS | 53.1 (1.4) | 29.0 (0.6) | 60.6 (2.3) | 36.7 (1.0) |
| CP CROCS | 51.0 (0.4) | 29.7 (1.0) | **67.5** (1.0) | **42.0** (6.5) |

Table 10: **Precision of $K$ retrieved representations, $v$, in the validation set of Chapman and PTB-XL, that are closest to the query when CROCS is trained with 10% of labelled data.** Results are shown for partial and exact matches of the attributes (# attribute matches) represented by the query and retrieved cardiac signals, and are averaged across five random seeds. Brackets indicate standard deviation and bold reflects the top-performing method. The strong performance of TP CROCS provides evidence in support of our CROCS framework.

| # attribute matches | Query | Chapman | | | PTB-XL | | |
| --- | --- | --- | --- | --- | --- | --- | --- |
| | | $K = 1$ | 5 | 10 | 1 | 5 | 10 |
| $\geq 1$ | DTC [38] | 71.9 (0.0) | 100.0 (0.0) | 100.0 (0.0) | 70.0 (0.0) | 100.0 (0.0) | 100.0 (0.0) |
| | TP CROCS | **91.3** (1.3) | 98.1 (3.8) | 100.0 (0.0) | **91.8** (3.4) | 99.5 (1.0) | 100.0 (0.0) |
| | CP CROCS | 89.4 (2.5) | 98.1 (3.8) | 100.0 (0.0) | 88.0 (6.2) | 100.0 (0.0) | 100.0 (0.0) |
| $\geq 2$ | DTC [38] | 25.0 (0.0) | 71.3 (1.3) | 89.0 (4.6) | 21.0 (0.0) | 50.1 (5.0) | 78.5 (8.0) |
| | TP CROCS | **53.8** (3.6) | 85.6 (7.0) | 92.5 (1.5) | **62.6** (12.7) | **91.3** (4.8) | 94.9 (1.6) |
| | CP CROCS | 51.3 (2.5) | **86.9** (7.5) | **93.8** (7.1) | 57.5 (5.2) | 90.0 (5.2) | **97.5** (3.2) |
| $= 3$ | DTC [38] | 3.1 (0.0) | 12.5 (0.0) | 21.9 (2.0) | 2.5 (0.0) | 12.0 (1.0) | 20.5 (4.0) |
| | TP CROCS | 11.3 (1.5) | **30.0** (3.2) | 40.6 (4.4) | 10.3 (3.2) | **28.7** (4.4) | 37.9 (6.4) |
| | CP CROCS | 11.3 (1.5) | 28.8 (7.0) | **43.8** (5.2) | 9.5 (1.9) | 27.5 (0.0) | **39.0** (5.2) |

We find that, in both the clustering and retrieval settings, our framework, CROCS, continues to generalize well and outperform the baseline methods. For example, on Chapman, CP CROCS achieves Acc(class) = 0.83, whereas KM EP, the next best baseline method, achieves Acc(class) = 0.65. This relative improvement also holds on the PTB-XL dataset. Overall, such a finding provides evidence that CROCS is relatively robust to the amount of labelled training data that are available and can thus be useful in realistic settings characterized by scarce, labelled data.

# H    Deploying clinical prototypes in the retrieval setting

## H.1    Chapman

In the main manuscript, we qualitatively evaluated the retrieval performance of a DTC-derived prototype and a clinical prototype on the Chapman dataset. In this section, we continue this evaluation however for a different query; a TP CROCS query, which reflects the average of representations associated with a set of patient attributes. In Fig. 7 (top row), we present the distributions of the Euclidean distance between the query and the representations in the validation set of Chapman. In Fig. 7 (bottom row), we illustrate the six cardiac signals that are closest to the query. We find that the query is closer to representations of the class attribute than to those of a different class attribute. This is evident by the long tail of distance values exhibited between representation with $SR$ and the query $\{SR, male, under \ 49\}$.

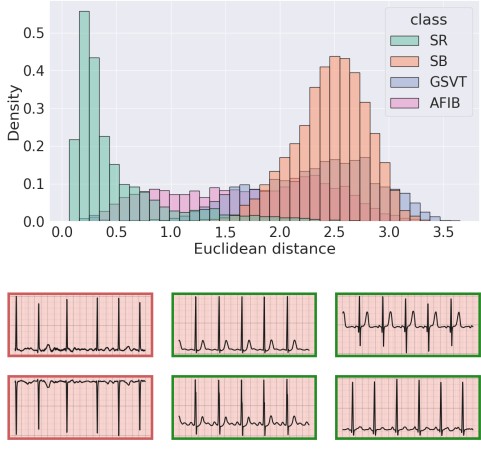

(a) TP CROCS query $\{SR, male, under \ 49\}$

Figure 7: **Qualitative retrieval performance of a TP CROCS query.** (**top row**) Euclidean distance from a query to representations in the validation set of Chapman. (**bottom row**) Six closest cardiac signals to each query. The query is associated with a set of patient attributes $\{disease, sex, age\}$. Retrieved cardiac signals with green borders indicate those whose class attribute matches that of the query. We see that the mean representation query is closer to representations of the same class $(SR)$ than to those of a different class, and thus retrieves relevant cardiac signals.

## H.2 PTB-XL

In this section, we continue our qualitative evaluation of the retrieval performance of various methods. In Fig. 8 (top row), we present the distributions of the Euclidean distance between the query (DTC-derived prototype or clinical prototype) and the representations in the validation set of PTB-XL. In Fig. 8 (bottom row), we illustrate the six cardiac signals that are closest to the respective prototypes. As with the results in the main manuscript, we find that the clinical prototype is closer to representations of the same class attribute than to those with a different class attribute. This is evident by the long tail of distance values exhibited between representation with MI and the clinical prototype {MI, female, over 95}. This behaviour, which is non-existent for the DTC-derived prototype, can explain the relatively improved retrieval performance of clinical prototypes. This is further supported by the retrieved cardiac signals (Fig. 8 bottom row) where the DTC-derived prototype and the clinical prototype retrieve relevant instances 0% and 50% of the time, respectively.

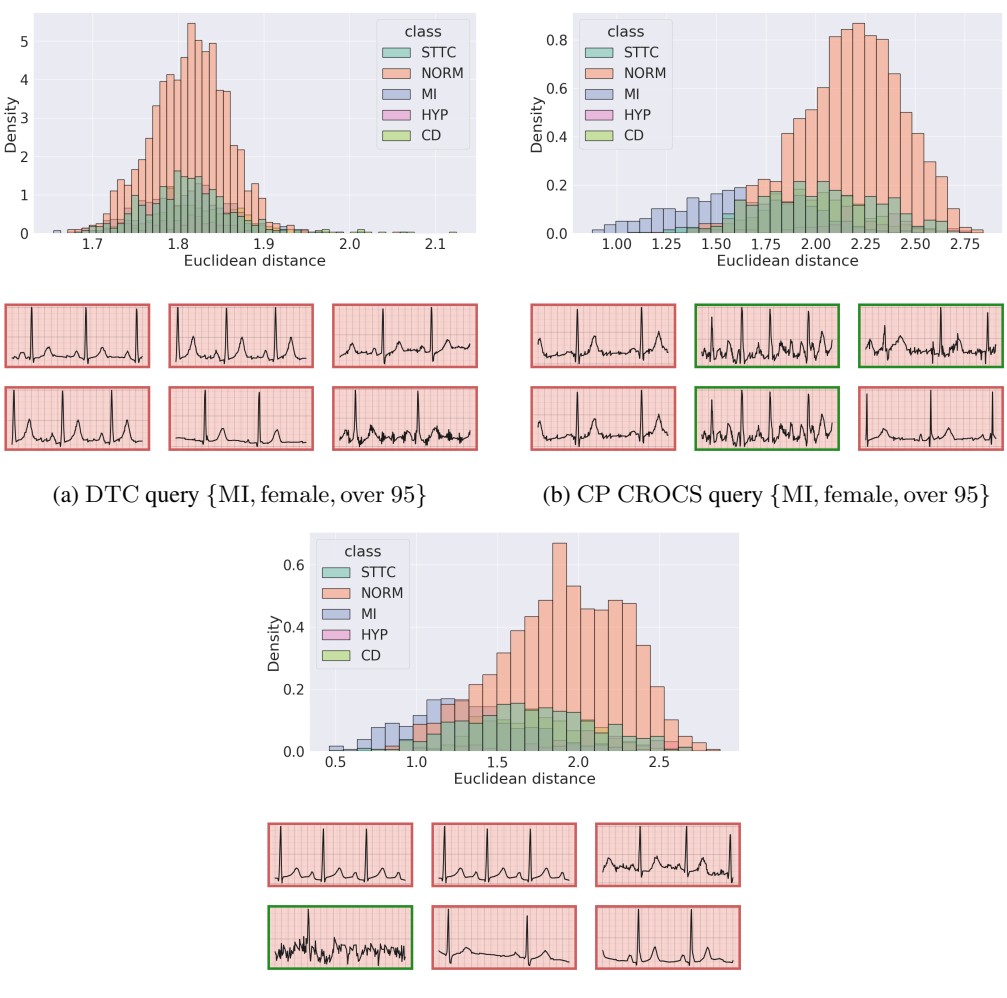

(a) DTC query {MI, female, over 95}

(b) CP CROCS query {MI, female, over 95}

(c) TP CROCS query {MI, female, over 95}

Figure 8: **Qualitative retrieval performance of two distinct queries.** (**top row**) Euclidean distance from a query (a) DTC (b) CP CROCS or (c) TP CROCS query, to representations in the validation set of PTB-XL. (**bottom row**) Six closest cardiac signals to each query. Each query is associated with a set of patient attributes {disease, sex, age}. Retrieved cardiac signals with green borders indicate those whose class attribute matches that of the query. We show that the clinical prototype is closer to representations of the same class (MI) and thus retrieve relevant cardiac signals.

# I  Investigating marginal impact of design choices

In this section, we quantify the marginal impact of the design choices of our CROCS framework on the retrieval performance. In Table 11, we present the precision of retrieved cardiac signals when evaluated based on both partial and exact matches of attributes also represented by the query. Each query is a clinical prototype that is learned in a variant of the CROCS framework. These variants are shown in Sec. 4 of the main manuscript. We find that clinical prototypes learned via our full framework ($\mathcal{L}_{NCE-soft} + \mathcal{L}_{reg}$) add value relative to those learned under the Hard assignment framework. For example, at $K = 1$, and when # attribute matches $\geq 2$, $\mathcal{L}_{NCE-hard}$, $\mathcal{L}_{NCE-soft}$ $\tau_\omega = \infty$, $\tau_\omega \neq \infty$, and $\mathcal{L}_{NCE-soft} + \mathcal{L}_{reg}$ achieve a precision of 27.5, 51.5, 58.5, and 63.0, respectively.

Table 11: **Marginal impact of design choices of CROCS on the precision of $K$ retrieved representations, $v$, in the validation set of Chapman and PTB-XL, that are closest to the query.** Results are shown for partial and exact matches of the attributes (# attribute matches) represented by the query and retrieved cardiac signals, and are averaged across five random seeds. Brackets indicate standard deviation and bold reflects the top-performing method.

| # attribute matches | Query | PTB-XL | | |
|---|---|---|---|---|
| | | $K = 1$ | 5 | 10 |
| $\geq 1$ | $\mathcal{L}_{NCE-hard}$ | 70.0 (3.9) | 100.0 (0.0) | 100.0 (0.0) |
| | $\mathcal{L}_{NCE-soft}$ | | | |
| | $\tau_\omega = \infty$ | 91.5 (2.0) | 100.0 (0.0) | 100.0 (0.0) |
| | $\tau_\omega \neq \infty$ | 88.0 (6.0) | 100.0 (0.0) | 100.0 (0.0) |
| | $+\mathcal{L}_{reg}$ | **92.5** (0.0) | 100.0 (0.0) | 100.0 (0.0) |
| $\geq 2$ | $\mathcal{L}_{NCE-hard}$ | 27.5 (3.9) | 67.5 (0.0) | 93.0 (4.0) |
| | $\mathcal{L}_{NCE-soft}$ | | | |
| | $\tau_\omega = \infty$ | 51.5 (2.0) | 94.5 (1.0) | 100.0 (0.0) |
| | $\tau_\omega \neq \infty$ | 58.5 (2.0) | 100.0 (0.0) | 100.0 (0.0) |
| | $+\mathcal{L}_{reg}$ | **63.0** (1.9) | 96.5 (2.0) | 99.5 (1.0) |
| $= 3$ | $\mathcal{L}_{NCE-hard}$ | 7.0 (1.0) | 16.0 (3.0) | 26.5 (4.6) |
| | $\mathcal{L}_{NCE-soft}$ | | | |
| | $\tau_\omega = \infty$ | 9.5 (1.0) | 30.5 (4.0) | 38.5 (3.0) |
| | $\tau_\omega \neq \infty$ | 12.5 (0.0) | 36.5 (2.0) | 43.5 (1.2) |
| | $+\mathcal{L}_{reg}$ | 12.5 (0.0) | 33.5 (3.0) | 43.0 (4.0) |

# J   Discovering attribute-specific features within clinical prototypes

We have shown that clinical prototypes can be deployed successfully for retrieval and clustering purposes while managing to capture relationships between attributes. In this section, we aim to quantify the relationship between clinical prototypes and explore their features further. In Fig. 9, we illustrate a matrix of the clinical prototypes ($M = 32$) along the rows and their corresponding features ($E = 128$) along the columns. By implementing the hierarchical agglomerative clustering (HAC) algorithm, we cluster these clinical prototypes and arrive at the dendrogram presented along the rows of Fig. 9. In addition to being correctly clustered according to class labels, they are also more similar to one another based on their attributes. This can be seen by the attribute combination descriptions in the right column. This finding supports our earlier claim that clinical prototypes do indeed capture relationships between attributes.

Motivated by recent work on disentangled representations, whereby representations can be factorized into multiple sub-groups each of which correspond to a particular abstraction, we chose to cluster the *features* of the clinical prototypes, resulting in the dendrogram presented along the columns of Fig. 9. The intuition is that by clustering we may discover attribute-specific feature subsets. We show that these features can indeed be clustered into three main groups, potentially coinciding with our pre-defined attributes. Such a process can improve the interpretability of clinical prototypes and lead to insights about how they can be further manipulated for retrieval purposes, for instance, by altering a subset of features.

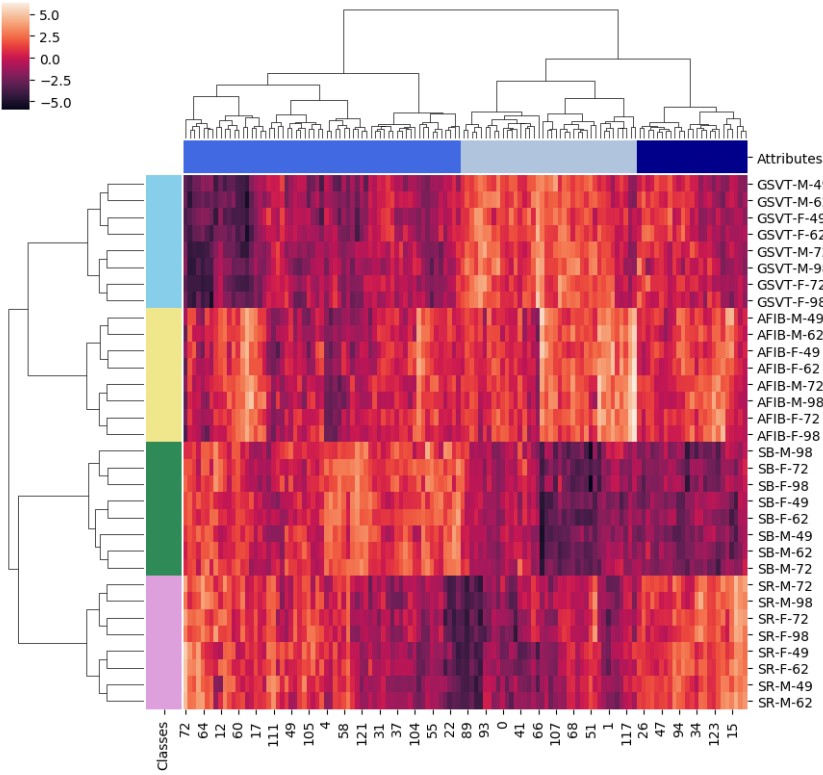

Figure 9: **Hierarchical agglomerative clustering of the clinical prototypes**. **(rows)** Clustering is performed using the 128-dimensional features resulting in 4 major clusters corresponding to the 4 classes. Clinical prototypes with similar attributes are also clustered together. The rows are labelled according to the attribute combination, $m$. **(columns)** Clustering is performed whereby each of the 128 *features* is treated as an instance, resulting in 3 major clusters which are hypothesized to correspond to the 3 attributes: class, sex, and age. This suggests that disentangled, attribute-specific features may have been learned.