# OpenReview forum: "CROCS: Clustering and Retrieval of Cardiac Signals Based on Patient Disease Class, Sex, and Age"
_NeurIPS.cc/2021/Conference — NeurIPS 2021 Poster_

### Official Review · Reviewer_mzV8 · 2021-07-16

**Rating:** 5
**Confidence:** 4

**Summary:**

Cardiac signals when unlabeled are difficult to query for given a set of attributes of interest such as disease class, age and sex.  The paper proposes a cluster-based retrieval system for this purpose. The algorithm proposed is of two-fold - (1) Learn a function to map a cardiac signal to an embedding closer in representation to a clinical prototype (embedding representative of a unique attribute set)   - this is done using a contrastive-learning framework assuming an initial-set of labeled data where cardiac-signals are annotated with class, age and sex and (2) Given a query attribute set - use its clinical prototype embedding to rank the  cardiac signals. Here the function from step 1 is used to get an embedding representation for cardiac signals and then their Euclidean distance from clinical prototypes is considered for ranking in the retrieval. The paper evaluates clustering and retrieval tasks on two datasets Chapman and PTB-XL.

**Limitations And Societal Impact:**

The paper discusses some of the limitations of the algorithm. Please find more detailed comments above to extend on this.

**Main Review:**

Briefly my overall review was primarily based on the following - the ideas presented in the paper are interesting to me, however I found that the setting considered is unrealistic and the evaluation provided is insufficient.  Please find my detailed review below -

There are several things to like about the paper,

Originality

(1) I liked the idea of extending multiple attribute information into unique clusters with their own clinical prototypes.

(2) I liked the regularization term to introduce semantic meaning - i.e prototypes representing similar attribute sets (e.g., adjacent age groups) should be closer to one another. The way this is modeled in equation 3 (using hamming distance of original attribute sets for supervision) is very interesting to me.

(3) Overall I found the paper easy to read (except the description on retrieval setting which I think can be illustrated with an algorithm and I think the intuition of the metric (in equation 5) should be discussed)

There are several shortcomings as well -

Originality

 (1) I think the setting considered is slightly unrealistic. Typically the number of unlabeled data is orders of magnitude higher than the labeled data in such clinical retrieval tasks. Also the unlabeled data can be subjected to distribution shifts as compared to the labeled data distribution. Owing to both these difficulties existing literature has not explored the direction of using supervised clustering  (as this paper did) - but rather tried weak-supervision or unsupervised techniques of clustering.  This changes the setting in several ways - (a) In the paper the retrieval evaluation is done on the validation dataset (20%) which is smaller than the training data (60%). This will change the retrieval metrics - meaning the metrics reported in the paper can be overestimating the actual performance (b) Supervised learning may not be an ideal framework any more and semi-supervised learning would need to be considered to address the above challenges.

(2) How will the algorithm perform as we decrease the labeled data and increase the unlabeled data? Can you provide retrieval and clustering metrics for the same and also compare this approach with semi-supervised/self-supervised clustering algorithms? (currently retrieval metrics are only compared against DTC a supervised learning method.)

(3) Although the application considering multiple patient attributes for retrieval is new - this comes with several limitations that needs to be discussed in the paper (a) the complexity of algorithm or the clusters considered will increase exponentially with increase in the attributes and the number of values taken by each attribute. This will also limit the amount of discretization that can be done in attributes such as age.  (b)  This will also mean assuming sufficient labeled data covering all these attribute information.

Quality/Clarity -

The retrieval metrics (equation 5) are not reflective of precision at K and are much relaxed versions - the metric is not average P@K across all M clusters (or queries) but rather average of the times P@K is non-zero across the M clusters. Can you provide a discussion on this metric?

I have several questions - can you please help by answering them?

(1) In figure 3 - Is this representative of the real data experiments or is this a cartoon to visualize the algorithm? If it is indeed a zoomed-in figure for experimental data - please provide more details. If it is not can you validate that the model has learnt semantically meaningful representations?

(2) In Figure 4 right image - points from following classes - HYP and NORM seem to be indistinguishable. So is a part of STTC and MI. Considering this the accuracy reported in Table 1 (76%) seems a little high especially considering HYP and NORM are majority classes.

(3) Does DTC in Table 1 really have a standard deviation of 15.0 in Accuracy and 20 in AMI? How is this possible - please provide an explanation?

Significance -

It is difficult to evaluate the significance of the paper considering insufficient evaluation and unrealistic setting as compared with existing work.


**Time Spent Reviewing:**

8-9 hrs

---

> ### Author Response · Authors · 2021-08-06
> **Response to Reviewer mzV8 - Round A Part 1**
>
> We would like to thank the reviewer for taking the time and effort to review our manuscript and for providing us with valuable feedback. We have addressed your comments below and modified the manuscript accordingly.
>
> **Framework Clarifications**
>
> We would like to clarify a potential misunderstanding of our proposed framework by the reviewer. It appears that the reviewer has suggested that our framework is a 'cluster-based retrieval system'. However, we believe this is misleading as it suggests that our framework is focused exclusively on retrieval. To clarify, our supervised contrastive learning framework allows for, and performs, both the tasks of clustering and retrieval separately from one another. Upon clarifying this note, our novelty relative to previous frameworks becomes more apparent. First, previous frameworks either perform clustering or retrieval and do NOT have the capability/flexibility of performing both tasks. In contrast, our framework can perform both clustering and retrieval. Second, previous frameworks which perform either clustering or retrieval do so based on a single patient attribute (e.g., disease class). In contrast, our framework can be applied to multiple patient attributes, and we have demonstrated this by exploiting the attributes of disease class, sex and age as exemplars.
>
> **Consideration of Realistic Setting**
>
> We agree with the reviewer's statement that, in many clinical scenarios, unlabelled data is more prevalent than its labelled counterpart. In paragraph 1 of the introduction, for example, we emphasize the increased prevalence of unlabelled instances. Distribution shift is also a problematic phenomenon that it exhibited in instances that are unlabelled relative to their labelled counterparts. Despite the validity of these statements, we would like to respectfully disagree with the reviewer's statement that our experimental setting is unrealistic. We describe the clinical motivation behind our setting below.
>
> Our experimental settings (clustering and retrieval) are driven by the two questions that we posed in Lines 24-26 in the manuscript, which, in turn, are clinically-motivated. If we were to focus on the first question posed, it revolves around stakeholders within healthcare who need a way to search for and retrieve relevant instances from a clinical database. This becomes difficult to achieve if such instances happen to be unlabelled. To address this challenge, we train on labelled data to learn clinical prototypes and use such prototypes to retrieve relevant unlabelled instances. Although the effect of the relative size of the labelled and unlabelled data on performance would be an interesting exploration (see Comment 2 below for new results in this proposed setting), we believe that it does not discount from how realistic / clinically-motivated the setting is. Although exploring distribution shift would also be an interesting segue, we believe that it is beyond the scope of our work.
>
> On the topic of semi-supervised learning, recent work has already explored the use of contrastive learning (as a network pre-training mechanism) with electrocardiogram signals in a semi-supervised setting (see Ref 20). In contrast, we are focusing on the task of supervised clustering and retrieval, both of which have a clinical motivation. We are happy to emphasize this clinical motivation in the camera-ready version of the manuscript.
>
> **Performance with Less Labelled Data**
>
> To address the reviewer's comment, we have now conducted all experiments in the manuscript with a reduced amount of labelled data. More precisely, we have reduced the amount of labelled training data 10-fold while keeping the amount of unlabelled data fixed. This should reflect the more realistic setting (less labelled data than unlabelled data) that the reviewer had suggested. In Tables 1 and 2, we present the results of these experiments in the clustering and retrieval settings, respectively. We are happy to incorporate these results into the camera-ready version of the manuscript.
>
> We find that, in both the clustering and retrieval settings, our framework, CROCS, continues to generalize well and outperform the baseline methods. For example, on Chapman, CP CROCS achieves Acc(class)=0.83, whereas KM EP, the next best baseline method, achieves Acc(class)=0.65. This relative improvement also holds on the PTB-XL dataset. Overall, such a finding provides evidence that CROCS is relatively robust to the amount of labelled training data that are available and can thus be useful in realistic settings characterized by scarce, labelled data.
>
> **Table 1a** - Clustering Results on Validation Set for Disease Class Attribute (**10% Labelled Training Data**)
>
> |          	|   Chapman   	|             	|    PTB-XL   	|             	|
> |----------	|:-----------:	|:-----------:	|:-----------:	|:-----------:	|
> |          	|     Acc     	|     AMI     	|     Acc     	|     AMI     	|
> | SeLA     	|      *      	|      *      	|   0.10 (0)  	| 0.02 (0.01) 	|
> | DC       	|   21.0 (0)  	|  0.5 (0.6)  	|   10.5 (0)  	|  4.3 (0.9)  	|
> | IIC      	|  27.2 (0.3) 	|  0.6 (0.01) 	|  24.3 (2.8) 	|  0.4 (0.07) 	|
> | KM Raw   	|  28.4 (1.2) 	|  0.3 (0.02) 	|      -      	|      -      	|
> | DTCR     	| 34.34 (0.9) 	|   3.27 (0)  	|  24.1 (0.5) 	|  0.8 (0.3)  	|
> | DTC      	|  46.3 (2.6) 	|  11.8 (2.2) 	|  48.4 (3.9) 	|  0.3 (0.5)  	|
> | KM EP    	| 64.87 (4.7) 	|  44.3 (3.4) 	| 45.06 (1.9) 	|  20.7 (1.3) 	|
> | KM CROCS 	|  71.1 (4.6) 	|  52.7 (2.3) 	|  47.7 (3.4) 	|  20.1 (0.9) 	|
> | TP CROCS 	|  75.5 (0.2) 	|  56.8 (0.5) 	|  47.3 (2.8) 	|  19.9 (1.4) 	|
> | CP CROCS 	|  **82.7 (0.4)** 	|**61.75 (0.8)** 	|  **71.38 (0)**  	|  **28.8 (0.8)** 	|
>
> *Algorithm could not be solved
>
> **Table 1b** - Clustering Results on Validation Set for Sex and Age Attributes (**10% Labelled Training Data**)
>
> |          	|   Chapman   	|             	|   PTB-XL   	|            	|
> |:--------:	|:-----------:	|:-----------:	|:----------:	|:----------:	|
> |          	|     Sex     	|     Age     	|     Sex    	|     Age    	|
> | DTCR     	|  52.2 (1.2) 	|  26.7 (0.3) 	| 51.7 (1.4) 	| 29.3 (1.2) 	|
> | DTC      	|  53.1 (0.6) 	| 26.8 (0.03) 	|  52.2 (0)  	| 33.5 (1.6) 	|
> | KM EP    	| 56.4 (0.06) 	|  30.2 (0.9) 	| 51.0 (0.7) 	| 37.1 (1.5) 	|
> | KM CROCS 	| **56.2 (0.06)** 	|  **30.9 (0.8)** 	| 51.1 (0.7) 	| 36.0 (1.1) 	|
> | TP CROCS 	|  53.1 (1.4) 	|  29.0 (0.6) 	| 60.6 (2.3) 	| 36.7 (1.0) 	|
> | CP CROCS 	|  51.0 (0.4) 	|  29.7 (1.0) 	| **67.5 (1.0)** 	| **42.0 (6.5)** 	|
>
> **Table 2** Retrieval Results on Validation Set (**10% Labelled Training Data**)
>
> | #   of attribute matches 	|   Query  	|   Chapman  	|             	|            	|   	|    PTB-XL   	|            	|            	|
> |:------------------------:	|:--------:	|:----------:	|:-----------:	|:----------:	|:-:	|:-----------:	|:----------:	|:----------:	|
> |                          	|          	|      1     	|      5      	|     10     	|   	|      1      	|      5     	|     10     	|
> |                          	|    DTC   	|  71.9 (0)  	|   100 (0)   	|   100 (0)  	|   	|   70.0 (0)  	|   100 (0)  	|   100 (0)  	|
> |        at least 1        	| TP CROCS 	| **91.3 (1.3)** 	|  98.1 (3.8) 	|   100 (0)  	|   	|  **91.8 (3.4)** 	| 99.5 (1.0) 	|   100 (0)  	|
> |                          	| CP CROCS 	| 89.4 (2.5) 	|  98.1 (3.8) 	|   100 (0)  	|   	|  88.0 (6.2) 	|   100 (0)  	|   100 (0)  	|
> |                          	|    DTC   	|   25 (0)   	| 71.25 (1.3) 	| 89.0 (4.6) 	|   	|    21 (0)   	| 50.1 (5.0) 	| 78.5 (8.0) 	|
> |        at least 2        	| TP CROCS 	| **53.8 (3.6)** 	|  85.6 (7.0) 	| 92.5 (1.5) 	|   	| **62.6 (12.7)** 	| **91.3 (4.8)** 	| 94.9 (1.6) 	|
> |                          	| CP CROCS 	| 51.3 (2.5) 	|  **86.9 (7.5)** 	| **93.8 (7.1)** 	|   	|  57.5 (5.2) 	|  90 (5.2)  	| **97.5 (3.2)** 	|
> |                          	|    DTC   	|   3.1 (0)  	|   12.5 (0)  	| 21.9 (2.0) 	|   	|   2.5 (0)   	|  12 (1.0)  	| 20.5 (4.0) 	|
> |           all 3          	| TP CROCS 	| 11.3 (1.5) 	|   **30 (3.2)**  	| 40.6 (4.4) 	|   	|  10.3 (3.2) 	| **28.7 (4.4)** 	| 37.9 (6.4) 	|
> |                          	| CP CROCS 	| 11.3 (1.5) 	|   28.8 (7)  	| **43.8 (5.2)** 	|   	|  9.5 (1.9)  	|  27.5 (0)  	|  **39 (5.2)**  	|
>
>
> **Framework Limitations**
>
> The reviewer has mentioned several valid limitations which we had framed as potential avenues for future work in the manuscript (see Lines 292-299). In short, this comes down to the discretization of the patient attributes and the scalability of our framework. Having sufficient coverage of such patient attributes, as the reviewer suggested, is an excellent point and we will include such limitations in the camera-ready version of the manuscript.
>
> **Retrieval Evaluation Metric**
>
> We introduce and describe Equation 5 in Lines 177-184 of the manuscript. As described there, Equation 5 is a variant of the Precision at K metric. Indeed, as the reviewer suggested, this is a relaxed version of the Precision at K metric which we can explicitly state in the modified version of the manuscript.
>
> The metric can be described as follows: it is the fraction of the clinical prototypes (or queries, more broadly) that retrieve a relevant instance, given that each prototype is allowed to retrieve K instances in total. The motivation for using this metric is as follows: we experimentally found that achieving an exact patient attribute match (number of attributes matches = 3) was quite difficult and using the traditional Precision at K metric would conceal subtle changes in performance. By using this variant of the metric, we could better evaluate our framework and compare it to baseline methods.
>
> **NOTE - our response is continued in Round A - Part 2**

---

> > ### Author Response · Authors · 2021-08-06
> > **Response to Reviewer mzV8 - Round A Part 2**
> >
> > **Figure 3 Clarity**
> >
> > The clinical prototypes shown in Figure 3 were generated based on experiments conducted on the PTB-XL dataset (see Ref [33]). In the camera-ready version of the manuscript, we will include the relevant details in the figure caption.
> >
> > In short, Figure 3 (left) is based on our framework trained with a soft contrastive loss and uniform weighting (as described in Lines 119-125). Figure 3 (centre) is based on our framework trained with a soft contrastive loss and modulated weighting (as described in Lines 126-139). Figure 3 (right) is based on our framework trained with a soft contrastive loss, modulated weighting, and the regularization term (with the latter described in Lines 143-158).
> >
> > **Figure 4 and Clustering Results**
> >
> > To clarify, the disease classes NORM and STTC contain the most instances, and would be described as the majority classes (as opposed to HYP). In fact, HYP and MI contain the fewest instances. In light of this, and the decent separability exhibited between the three major classes, NORM, STTC, and CD, we believe the accuracy score of 76\% seems reasonable.
> >
> > **Standard Deviation of DTC**
> >
> > Yes, we have confirmed that the standard deviation of the accuracy of DTC when evaluated for the disease class on the Chapman dataset is indeed 15. Recall that all of our experiments are conducted across five random seeds. This involves five random network initializations and random orders with which the data are presented to the network. Such a high standard deviation suggests that DTC is quite sensitive to these initializations and data orderings.

---

### Official Review · Reviewer_2o6W · 2021-07-16

**Rating:** 8
**Confidence:** 4

**Summary:**

This paper proposes a clustering and information retrieval algorithm for unannotated large databases of physiological signals (in this case ECG signals). The method is based on a contrastive learning approach, which has recently been suggested for semi-supervised learning on ECG data.
The proposed approaches are evaluated on two large ECG datasets and compared with state-of-the-art approaches.


**Limitations And Societal Impact:**

The main concerns are the following:
1.	The explanation of using soft assignment instead of har-assignment is discussed in the description of the methodology, but it is unclear to me whether the authors have reported an experiment allowing to prove their conjecture.
2.	The authors should discuss the fact that CP CROCS seem to be underperforming the other clustering approaches based on the age attribute. This result is quite strange and an outlier, but it would be interesting to have a discussion on a possible explanation of this phenomenon.
3.	Could the authors discuss the main difference of the tow datasets, and especially as depicted in figure 4 why the separability of the classes is so much lower on PTB-XL compared to Chapman
4.	IN terms of quality of the information retrieval, cold the authors comment why evaluating soft assignment (and performance if only one attribute is correct) is clinical significant? It seems to me quite important that pathology and should be well retrieved, extracting an example of AF when asking for Normal rhythm, would have higher impact than a mistake on age or sex
5.	Can the authors discuss whether age and sex are good attributes for ECG signals? Heart Rate variability is affected by age, but I am not aware of morphological changes in the ECG signals due to age (or even sex)? Figure 3 seem to be indicating that the representation of the ECG signal is not that influenced by either age or sec, although there seem to be a trend or evolution with age, and separation of sex on the proposed full framework.
6.	The order in sections 2 related work and 3 background should be consistent clustering then IR.
7.	The acronym HER is not introduced properly, that is at the first appearance of the term


**Main Review:**

The paper is really innovative with two new applications (clustering and information retrieval) based on a recently suggested representation learning approaches.
The research is quite significant, given the availability of large databases of EHR data, containing clinical physiological signals (for instance the MIMIC III database). Applying such approach on such a dataset, with many meta-data available, could help uncover specific clinical protoytpes of physiological signlas and be of the utmost importance.
The paper is clearly written and of good quality. The proposed approach is tested in several settings, using two available large sets of ECG signals, and the technique performance is compared to other state-of-the-art approaches.


**Time Spent Reviewing:**

5

---

> ### Author Response · Authors · 2021-08-06
> **Response to Reviewer 2o6W - Round A**
>
> We would like to thank the reviewer for taking the time and effort to review our manuscript and for providing us with valuable feedback. We have addressed your comments below and modified the manuscript accordingly.
>
> **Soft Assignment vs. Hard Assignment**
>
> When making the design choice to transition from a hard assignment to a soft assignment, we were primarily motivated by the intuition outlined in the manuscript (Lines 112-114). However, we had also conducted experiments offline (not included in manuscript) that demonstrated the inferiority of the hard assignment approach relative to its soft assignment counterpart. As a result, we opted for the latter approach throughout the manuscript.
>
> **CP CROCS and Age Attribute**
>
> We hypothesize that the relatively poorer performance of CP CROCS for the age attribute on the PTB-XL dataset could be due to the learning of clinical prototypes that are more discriminative along the dimensions of other patient attributes (e.g., disease class and sex). This, in turn, could be due to the strength of the modulated attraction (introduced in Line 126).
>
> **Clarification of Datasets**
>
> The two datasets, Chapman and PTB-XL, although reflect the same data modality (ECG) and comprise similar patient attributes (disease class, sex, and age), differ in various ways. For example, Chapman and PTB-XL consist of 4 and 5 disease classes, respectively. Figure 4 (right) indicates that the disease classes are less separable than those in Figure 4 (left). This could be due to the disease classes being more similar to one another, the amount of data in each of the disease classes (note that there are few instances within MI class), and the network architecture used. We leave the exploration of more complex network architectures to future work.
>
> **Retrieval Evaluation Intuition**
>
> Ideally, a retrieved instance would match all of the query's attributes (disease class, sex, and age). In our context, we have broken down the evaluation of the retrieval setting based on the number of attribute matches. This is done for two reasons. First, this allows us to evaluate our framework at a more granular level and thus detect subtle changes in performance. Second, and from a clinical perspective, the relative importance of each of these attributes arguably depends on the clinical context. In other words, the pathology (i.e., disease class) might be most important, as the reviewer suggested, for a cardiologist diagnosing a disease. However, the age variable might be most important for a pharmaceutical company looking to recruit patients into a clinical trial that satisfy an age group criteria, or if they are looking to stratify the outcome of an analysis based on age.
>
> **Sex and Age from ECG**
>
> There has been some recent work looking to exploit the electrocardiogram alongside deep learning to estimate sex and age (https://www.ahajournals.org/doi/full/10.1161/CIRCEP.119.007284). Although promising, their system was trained on around 500,000 patients, significantly more than those available in publicly-available datasets (around 20,000). As such, it is likely that there is less of a signal in the Chapman and PTB-XL datasets to delineate ECGs based on sex and age. Having said that, our framework is agnostic to the patient attributes that are used. Our primary motivation for using such attributes (disease class, sex, and age) was based on their availability in the publicly-available datasets.
>
> **Ordering of Sections**
>
> We remained consistent with the ordering of the sections. For example, in the Related Work, Background, Experimental Design, and Experimental Results sections, we mentioned clustering before retrieval.
>
> **Acronyms**
>
> In the camera-ready version of the manuscript, we will introduce the electronic health record (EHR) acronym, and any other acronym, upon first appearance.

---

> > ### Comment · Reviewer_2o6W · 2021-08-25
> > **Comments to the authors' response**
> >
> > I thank the authors for their response.
> > I believe that the difference in separability of the classes in the two tested databases is worth discussing and exploring. I am not convinced that this could be explained by classes being more similar in one database than the other. I am really interested in finding an explanation in this phenomenon, and wonder whether this could be done on the population of each database. Does any of this database contain multiple recordings from the same patient?
> > I understand that the method is agnostic to the attributes, and could be trained on any set of attributes, but I also understand that most commonly available attributes are indeed gender and age. Estimation of age from ECG signals is indeed (as pointed out by the authors in their response) gaining attention, but I am curious to understand what the models are looking at the signal to estimate the subject’s age, and what could be the physiological phenomenon that could explain alterations of the ECG signal. Is it morphological changes or has it to do with Heart Rate variation (as it is known that respiratory sinus arrhythmia is attenuating over the years)
> > Regarding the rating, I still believe this paper is worthy of acceptance to this conference.

---

### Official Review · Reviewer_vkP4 · 2021-07-20

**Rating:** 7
**Confidence:** 4

**Summary:**

This work presents a framework for information retrieval and clustering of patients based on their ECG’s. To do this, the authors framed the problem as a supervised contrastive learning problem – this is a very hot topic in machine learning for healthcare and I believe the framework and experimental design they propose is technically appropriate.

**Limitations And Societal Impact:**

The authors acknowledge several key limitations, mostly the result of the data that were available to them. I think the authors should discuss several other key limitations: 1.) in the evaluation of the methods, particularly a.) the re-implementation of prior methods, and b.) clinical evaluation of the information retrieval task. It should be straightforward to test whether a clinician believes the information retrieval matches their training (by showing a matched ECG and a further ECG and testing whether they identify the matched ECG). If they do not, these examples can be further explored – either the clinician is wrong, or the model could be improved. 2.) the lack of external validation or transfer to alternate datasets (e.g. training on Chapman and application to PTB-XL - even if only for Age, Sex)

This work has a potential to contribute to important societal impacts, particularly around fair machine learning for health. It may allow researchers to study underrepresented groups that are not clearly defined by binary categorizations of race, sex etc. but will need to be further studied to demonstrate this.


**Main Review:**

Overall, my opinion was that the work was technically sound, and my primary concerns are about novelty and the chosen benchmarks for comparison. If these two points can be clarified I believe this can be a suitable paper for this venue.

I had trouble differentiating the authors technical contribution and from prior work. I believe the authors have substantially built upon past work and have novel contributions but they should make it more clear what their key technical contributions are. I believe this is likely the key major point to evaluate this work for a venue like NeurIPS.

In line with this comment – the authors may want to break out a section from the clinical representation learning to include more of the recent contrastive learning papers.

Figure 3 is confusing - what data were used? is this a result or part of the methods? (simulated?). The legend also isn't explained (e.g. sttc).

I believe there are better metrics to show in table 1 (e.g. traditional classification metrics + confidence intervals).

The performance of examples such as SeLA, DC and IIC is so bad it’s likely they were not appropriately adapted to the problem – can this be explained? Are there better apples to apples comparisons? This is especially true given that with the supervised component, we know there are FDA approved devices with arrhythmia classification (AliveCor). I'm not convinced these are the best examples to compare to.

Why were methods removed transitioning from table 1 to table 2.

Minor Comments:
I found it strange that the information retrieval task as introduced first throughout, but then the clustering results were shown first.

I think you need to be careful about using the term clinical prototype, as it has a collision with the medical devices space. Given this work is using ECG data this will be confusing to the clinical audience. I admit, this may be a subjective preference as I’ve never liked the term cluster prototype, because it changes the traditional definition of the word prototype (a first instance other are copied from).



**Time Spent Reviewing:**

4

---

> ### Author Response · Authors · 2021-08-06
> **Response to Reviewer vkP4 - Round A**
>
> We would like to thank the reviewer for taking the time and effort to review our manuscript and for providing us with valuable feedback. We have addressed your comments below and modified the manuscript accordingly.
>
> **Contributions**
>
> Our contributions relative to previous work are twofold and can be summarized as follows. **First**, previous frameworks either perform clustering or retrieval and do NOT have the capability/flexibility of performing both tasks. In contrast, our framework can perform both clustering and retrieval. **Second**, previous frameworks which perform either clustering or retrieval do so based on a single patient attribute (e.g., disease class). In contrast, our framework can be applied to multiple patient attributes, and we have demonstrated this by exploiting the attributes of disease class, sex and age as exemplars.
>
> **Additional Related Work**
>
> We are more than happy to branch out the 'clinical representation learning' subsection within the Related Work section to include work on contrastive learning. As the reviewer has correctly pointed out, there has been recent relevant work on contrastive learning. Although we have mentioned the two contrastive learning papers (Refs 20 and 21) that are most relevant to our work, we will include further papers to reflect this rich domain. Such papers will include SimCLR (http://proceedings.mlr.press/v119/chen20j.html), BYOL (https://arxiv.org/abs/2006.07733), and https://arxiv.org/abs/2010.00747. We are happy to include additional papers that the reviewer believes are relevant.
>
> **Figure 3 Clarity**
>
> The clinical prototypes shown in Figure 3 were generated based on experiments conducted on the PTB-XL dataset (see Ref [33]). Although Figure 3 is based on such experiments, we decided to include it in the Methods section in order to communicate the motivation behind our system's design choices. Specifically, it provides the motivation behind our proposed (1) modulated attraction mechanism and (2) regularization term.
>
> As for the legend, it reflects the patient attributes (disease class, sex, and age groups) that each clinical prototype is associated with. In the camera-ready version of the manuscript, we will describe the legend in more detail and explain how the figure informed our design choices.
>
> **Baseline Methods**
>
> We believe that the relatively poorer performance of SeLA, DC, and IIC can be explained by the fact that they are inherently unsupervised methods. In comparison, our proposed framework is supervised (since it explicitly exploits patient attributes during the learning process) and therefore outperformed the aforementioned baseline methods. We further compared our approach to supervised clustering baseline methods (i.e., DTC and KM EP) which are noticeably better than their unsupervised counterparts, as would be expected.
>
> As for comparing our approach to FDA-approved cardiac arrhythmia classification algorithms (or any other exclusively supervised algorithm), we believe this would be an unfair comparison for the following reason. Our approach jointly optimizes for multiple patient attributes (disease class, sex, and age) whereas cardiac arrhythmia classification algorithms are optimizing for a single attribute (i.e., disease class). Having said that, and with these caveats in mind, we are happy to include such a supervised baseline (which can be found in the Supplementary Material associated with Ref 20, for example).
>
> **Table 1 and Table 2 Baseline Methods**
>
> Table 1 reflects the results of experiments conducted in the clustering setting whereas Table 2 reflects the results of experiments conducted in the retrieval setting. Several methods shown in Table 1 are clustering methods which do not trivially extend to the retrieval setting. As such, they were not included in Table 2. However, as shown in the manuscript, we adapted the DTC method to operate in both the clustering and retrieval settings.
>
> **Ordering of Sections**
>
> We remained consistent with the ordering of the sections. For example, in the Related Work, Background, Experimental Design, and Experimental Results sections, we mentioned clustering before retrieval. We do, however, spot a reversal of that order in the introduction. If the reviewer is referring to this, then we are happy to modify the order to improve the coherence of the paper.
>
> **Prototype Terminology**
>
> We thank the reviewer for bringing this to our attention. Indeed the term 'prototype' may be difficult to disambiguate. However, in our context, we have been careful to explicitly state that a clinical prototype is an embedding. For example, this can be seen when we first use the phrase 'clinical prototype' in Lines 40-41.
>
> **Framework Limitations**
>
> As for the limitations of the manuscript, we will be incorporating the reviewer's suggestions into the camera-ready version of the manuscript. We also agree with the reviewer that quantifying the clinical utility of an algorithm is an important component of algorithmic design within healthcare. We will emphasize this importance in the discussion and future work section of our camera-ready manuscript.
>
> **Impact**
>
> We thank the reviewer for emphasizing the potential positive impact this research may have on a multitude of stakeholders within healthcare. We believe that it could inform researchers working on fairness and thus better serve patients from underrepresented backgrounds.

---

### Official Review · Reviewer_XjBr · 2021-07-21

**Rating:** 5
**Confidence:** 4

**Summary:**

Paper presents for retrieving cardiac signal data corresponding to a patient described  by a set (3) of attributes (Sex, age, disease type). This is built on a model constructed by clustering such cardiac signals representations and learning their corresponding patient attributes.
	Although well written, the paper suffers, in our opinion, from targeting a very simplified problem (number of attributes, attribute sampling) that lends it to building a bespoke solution that overfits the used data and thus overperforms other methods.  The novelty was not clearly apparent to us from other clustering/retrieval methods.

**Ethical Concerns:**

Even if nominative information is not used, could have explained any ethical rules governing establishments or country where data is sourced from regarding use of medical or personal information records and the legislated limitations.

**Limitations And Societal Impact:**

No specific ones come to mind.

**Main Review:**

- "by the rapid growth of large-scale clinical databases and the increased prevalence of unlabeled instances; those for which patient attribute information is unavailable."
	- This covers a multitude of possible cases (some diagnostics data missing, variability in acquired diagnostics data from different origins (equipment, institutions, countries etc.) from different patients, missing patient data). A few examples of the ones you specifically target would be welcome at this stage.
- "Such methods, however, are exclusively unsupervised; they do not exploit patient attribute information."
	- Clustering approaches are unsupervised by design and meant to be. What do they cluster if they do not exploit patient attribute information? This part needs clarification. At this stage should define what data you are using and how you are segmenting it (patient attributes (disease, age, sex etc., other data (diagnostics?)).
- Figure 2:
	- Need to explain what the attributes are, what sinus rhythm/Atrial fib./ Bradycardia are (disease, disease class etc. for non domain expert. (later one learns of 5/6 disease classes used).
	- Is that a "dual" repulsion line towards the triangles of atrial fib.?
	- For the Similarity probability mass, can you provide the domain knowledge supporting it? Why/Is gender/sex that more indicative of cardiac signal discrepancies? How do the triangle/+ symbols used relate to the age, disease class attributes?

Notes:
- Please introduce your acronyms at first use to avoid confusions.
    - Example: Electronic Health Record (EHR) data.
- "... a centroid groups together instances that share some similarities.":
    - Isn't the centroid rather a single point used to represent a group of instances that share some similarities?

After rebuttal:
-----------------
Thanks to the authors for their responses.
After reading the rebuttals, many unclear points were explained.
I am still not convinced by the adequacy of the chosen experiment attributes and the generalization power of the approach.
Therefore, I am raising my evaluation to "Marginally below the acceptance threshold".
Regards.

**Time Spent Reviewing:**

5

---

> ### Author Response · Authors · 2021-08-06
> **Response to Reviewer XjBr - Round A**
>
> We would like to thank the reviewer for taking the time and effort to review our manuscript and for providing us with valuable feedback. We have addressed your comments below and modified the manuscript accordingly.
>
> **Framework Clarifications**
>
> We would like to clarify a potential misunderstanding of our proposed framework by the reviewer. It appears that the reviewer has suggested that our model is initially constructed by clustering and then applied for the retrieval of cardiac signals. However, we believe this is misleading since, in practice, our supervised contrastive learning framework actually allows for, and performs, both the tasks of clustering and retrieval separately from one another. Upon clarifying this note, our novelty relative to previous frameworks becomes more apparent. **First**, previous frameworks either perform clustering or retrieval and do NOT have the capability/flexibility of performing both tasks. In contrast, our framework can perform both clustering and retrieval. **Second**, previous frameworks which perform either clustering or retrieval do so based on a single patient attribute (e.g., disease class). In contrast, our framework can be applied to multiple patient attributes, and we have demonstrated this by exploiting the attributes of disease class, sex and age as exemplars. Although we had chosen these three attributes due to their availability in publicly-available cardiac datasets, our framework is flexible to incorporate additional attributes of interest.
>
> **Mentioning Attribute Information**
>
> At the beginning of paragraph 1 in the introduction, we explicitly mention the patient attributes that we are interested in working with (i.e., disease class, sex, and age). We are happy to reiterate these attributes at the end of the paragraph if the reviewer believes this will improve the clarity of the introduction.
>
> **Supervised Clustering and Data Clarifications**
>
> Indeed, there exist several clustering methods in the literature which are unsupervised (e.g., K-means). In this context, and depending on the algorithm, instances that are more 'similar' to one another tend to be grouped closer to one another. This notion of 'similarity', however, is agnostic to explicit labels (e.g., patient attributes). Therefore, the labels associated with the clusters are unknown. In contrast, supervised clustering, with a rich literature, involves the exploitation of a label of some sort during the learning process. We believe that our work falls under this latter category given our exploitation of patient attributes during the learning process.
>
> As for defining the data that we use, we explicitly mention, at the beginning of paragraph 3 in the introduction, that our data modality of interest is the electrocardiogram (ECG) and our labels of interest are the patient attributes (i.e., disease class, sex, and age). An in-depth description of the data is provided in Section 5 of the manuscript with further details in Appendix A.
>
> **Figure 2 Clarity**
>
> To improve clarity, we will modify Figure 2 in the camera-ready version of the manuscript as follows. (a) Add repulsion line to all atrial fibrillation symbols. (b) Tag 'sinus rhythm', 'atril fib', and 'bradycardia' with the following disease class labels 'disease class 1', 'disease class 2', etc. (c) Include additional prototypes to reflect the patient attribute of 'age'. It is currently omitted to avoid clutter.
>
> As for the similarity probability mass, it reflects the desired distribution of similarities between the instance representation and the clinical prototypes that we would like to achieve in the particular example shown. The intuition is that the representation of an instance associated with attributes (Sinus Rhythm, Male, Under 25) should be most similar to the clinical prototype that reflects the exact same set of attributes, and less similar to the remaining clinical prototypes. In this way, we learn clinical prototypes that are attribute-specific. Our omission of the age attribute in Figure 2 (to avoid clutter) may have created some confusion. As mentioned above, we will modify Figure 2 to avoid this potential confusion.
>
> **Acronyms**
>
> In the camera-ready version of the manuscript, we will introduce the electronic health record (EHR) acronym, and any other acronym, upon first appearance.
>
> **Centroids**
>
> Yes, the reviewer's suggested definition of a centroid is accurate and, we believe, aligns well with our statement in the manuscript. If this remains unclear, we are happy to modify our statement to reflect the reviewer's suggestion.

---

### Comment · Area_Chair_69T2 · 2021-08-19
**Discussion Paper1691**

Thanks to the reviewers for these in-depth comments. The authors have provided quite thoughtful rebuttals for each of the reviewers' comments. Given the variance in evaluation and scores, can I please ask reviewers to read through the other reviews and the authors' rebuttal to these reviews and also the rebuttal to your own review. Please respond to the authors' comments ad state whether you agree with the discussion points that they have raised. Given the rebuttal and the comments raised by the other reviewers, are you satisfied with the responses? Given these elements, would you revise your score?

---

> ### Comment · Reviewer_XjBr · 2021-08-23
> **Assessment after authors' rebuttals**
>
> After reading the authors rebuttals, many unclear points were explained.
> I am still not convinced by the adequacy of the chosen experiment attributes and the generalization power of the approach.
> Therefore, I am raising my evaluation to "Marginally below the acceptance threshold".
> Regards.

---

### Decision · Program_Chairs · 2021-09-27

**Decision:**

Accept (Poster)

**Comment:**

This presents a well written paper which presents a framework for information retrieval and clustering of patients based on unannotated ECG/ physiological data . In discussion, there were mixed reviews with regards to novelty, experimental evaluation and baseline comparisons to related work. However, after evaluating the paper and the discussion and reviewers' comments,  I agree that the proposed approach is presents a supervised technique for learning a compact representation which respect the attributes (or weak labelling) available. Such a representation could be used for other task for which access to a large database of (strong) annotated data is quite difficult. This is a non-trivial task in the healthcare domain and highlights an important application of ML for real-world impact, which is potentially generalisable to other data settings. Although there were concerns around over-fitting this model to a very stringent problem, the novelty of the approach and problem formulation from an ML perspective far outweighs any potential downsides. As both the authors and one of the reviewers highlights, this work may have a potentially positive impact on multitude of stakeholders within healthcare and could inform researchers working on fairness and thus better serve patients from underrepresented backgrounds.